# Exogenous Distribution Learning for Causal Bayesian Optimization

## Abstract

Maximizing a target variable as an operational objective within a structural causal model is a fundamental problem. Causal Bayesian Optimization (CBO) approaches typically achieve this either by performing interventions that modify the causal structure to increase the reward or by introducing action nodes to endogenous variables, thereby adjusting the data-generating mechanisms to meet the objective. In this paper, we propose a novel method that learns the distribution of exogenous variables-an aspect often ignored or marginalized through expectation in existing CBO frameworks. By modeling the exogenous distribution, we enhance the approximation fidelity of the data-generating structural causal models (SCMs) used in surrogate models, which are commonly trained on limited observational data. Furthermore, the ability to recover exogenous variables enables the application of our approach to more general causal structures beyond the confines of Additive Noise Models (ANMs) and single-mode Gaussian, allowing the use of more expressive priors for context noise. We incorporate the learned exogenous distribution into a new CBO method, demonstrating its advantages across diverse datasets and application scenarios.

## 1 Introduction

Bayesian Optimization (BO) is widely applied in domains such as automated industrial processes, drug discovery, and synthetic biology, where the objective is to optimize black-box functions (Močkus, 1975; Astudillo & Frazier, 2019; Garnett, 2023; Frazier, 2018). In many real-world scenarios, structural knowledge of the unknown objective function is available and can be exploited to enhance the efficiency of BO. Causal Bayesian Optimization (CBO) has been developed to incorporate such structural information (Aglietti et al., 2020; 2021; Sussex et al., 2023; Gultchin et al., 2023). CBO integrates principles from causal inference, uncertainty quantification, and sequential decision-making. Unlike traditional BO, which assumes independence among input variables, CBO accounts for known causal relationships among them (Aglietti et al., 2020). This framework has been successfully applied to optimize medical and ecological interventions (Aglietti et al., 2020; 2021), among other applications.

### 1.1 Approach and Contributions

In this paper, we propose a novel method called *EXogenous distribution learning augmented Causal Bayesian Optimization* (EXCBO). Given observational data from a structural causal model (SCM Pearl (2009; 1995)), our method recovers the exogenous variable corresponding to each endogenous node using an encoder-decoder framework, as illustrated in Figure 2. The recovered exogenous variable distribution is then modeled using a flexible density estimator, such as a Gaussian Mixture Model. This learned distribution significantly enhances the surrogate model's approximation of the underlying SCM, as shown in Figure 1.

Unlike existing CBO approaches (Aglietti et al., 2020; 2021; Sussex et al., 2023), which are typically confined to Additive Noise Models (ANMs Hoyer et al. (2008)), our method generalizes CBO to broader classes of causal models. By enabling the recovery of exogenous variables and their distributions, our surrogate model provides improved accuracy and flexibility for causal inference in the CBO update process.

The contributions of this work are as follows:

- We introduce a method for recovering the exogenous noise variable of each endogenous node in an SCM using observational data, which enables our model to capture *multimodal exogenous distributions*.
- This flexible approach to learning exogenous distributions allows our CBO framework to extend naturally to general causal models beyond the limitations of ANMs.
- We present a theoretical investigation of exogenous variable recovery through the proof of counterfactual identification, and we further analyze the regret bounds of the proposed algorithm.
- We conduct extensive experiments to evaluate the impact of exogenous distribution learning and demonstrate the practical advantages of EXCBO through applications such as epidemic model calibration, COVID-19 testing, and real-world planktonic predator–prey problem, etc.

The remainder of the paper is organized as follows. Section 2 reviews background and related work. Section 3 introduces the problem setup and outlines our proposed CBO framework. Section 4 presents the method for recovering exogenous variables. The proposed algorithm, EXCBO, is detailed in Section 5, followed by regret analysis in Section 6. Experimental results are presented in Section 7, and the paper concludes in Section 8.

## 2 BACKGROUND

We provide a brief overview of SCMs, intervention mechanisms, and CBO in this section.

### 2.1 STRUCTURAL CAUSAL MODEL

An SCM is denoted by $\mathcal{M} = (\mathcal{G}, \mathbf{F}, \mathbf{V}, \mathbf{U})$, where $\mathcal{G}$ is a directed acyclic graph (DAG), $\mathbf{F} = \{f_i\}_{i=0}^{d}$ represents the $d + 1$ structural mechanisms, $\mathbf{V}$ denotes the set of endogenous variables, and $\mathbf{U}$ the set of exogenous (background) variables. The generation of the $i$th endogenous variable follows

$$X_i = f_i(\mathbf{Z}_i, U_i); \ \mathbf{Z}_i = \mathbf{pa}(i), \ U_i \sim p(U_i), \text{ for } i \in [d]. \tag{1}$$

Here, $[d] = \{0, 1, \ldots, d\}$, and $X_i$ refers to both the variable and its corresponding node in $\mathcal{G}$. The set $\mathbf{pa}(i)$ denotes the parents of node $i$, while $\mathbf{ch}(i)$ refers to its children. We assume $U_i \perp\!\!\!\perp \mathbf{Z}_i$ and $U_i \perp\!\!\!\perp U_j$ for all $i \neq j$. Each $f_i$ is a mapping from $\mathbb{R}^{|\mathbf{pa}(i)|+1}$ to $\mathbb{R}$. The domains of $X_i$, $\mathbf{Z}_i$, and $U_i$ are denoted by $\mathcal{X}_i$, $\mathcal{Z}_i$, and $\mathcal{U}_i$, respectively. Additionally, we assume that the expectation $\mathbb{E}[X_i]$ exists for all $i \in [d]$. Most existing CBO approaches (Aglietti et al., 2020; 2021; Sussex et al., 2023) typically assume an Additive Noise Model (ANM Hoyer et al. (2008)) for exogenous variables, where $X_i = f_i(\mathbf{Z}_i) + U_i$ with $U_i \sim \mathcal{N}(0, 1)$.

### 2.2 INTERVENTION

In an SCM $\mathcal{M}$, let $\mathbf{I} \subset \mathbf{V}$ be a set of endogenous variables targeted for intervention. The post-intervention structural mechanisms are represented by $\mathbf{F}_x = \{f_i \mid X_i \notin \mathbf{I}\} \cup \{f_j \mid X_j \in \mathbf{I}\}$. A hard intervention replaces the mechanism for each $X_j \in \mathbf{I}$ with a constant value, resulting in $\mathbf{F}_x = \{f_i \mid X_i \notin \mathbf{I}\} \cup \{f_j := \alpha_j \mid X_j \in \mathbf{I}\}$, where $\boldsymbol{\alpha}$ is the realized value of the intervened variables. This corresponds to Pearl's do-operation (Pearl, 2009), denoted as $do(\mathbf{X_I} := \boldsymbol{\alpha})$, which alters $\mathcal{M}$ to a new model $\mathcal{M}_{\boldsymbol{\alpha}}$ by severing the dependencies between each $X_j$ and its parents.

This paper focuses on soft (or imperfect) interventions (Peters et al., 2017). Following the Model-based CBO framework (Sussex et al., 2023), we associate each endogenous variable with an action variable, modifying the mechanisms as $\mathbf{F}_x = \{f_i \mid X_i \notin \mathbf{I}\} \cup \{f_j := f_j(\mathbf{Z}_j, \mathbf{A}_j, U_j) \mid X_j \in \mathbf{I}\}$, where $\mathbf{Z}_j = \mathbf{pa}(j)$. Under soft intervention, the data-generating mechanism becomes

$$X_i = \begin{cases} f_i(\mathbf{Z}_i, U_i), & \text{if } X_i \notin \mathbf{I} \\ f_i(\mathbf{Z}_i, \mathbf{A}_i, U_i), & \text{if } X_i \in \mathbf{I} \end{cases}, \tag{2}$$

where $\mathbf{A}_i$ is a continuous action variable set associated with $X_i$ and takes values in $\mathcal{A}_i$. The soft intervention is represented using Pearl's notation as $do(\mathbf{X_I} := \mathbf{f}(\mathbf{Z_I}, \mathbf{A}, U_\mathbf{I}))$.

### 2.3 FUNCTION NETWORK BAYESIAN OPTIMIZATION

Function Network BO (FNBO Astudillo & Frazier (2021a; 2019)) operates under similar assumptions as CBO, where the functional structure is known but the specific parameterizations are not. FNBO applies soft interventions and employs an expected improvement (EI) acquisition function to guide the selection of actions. However, FNBO assumes a noiseless environment, which may limit its applicability in practical settings. Both FNBO and CBO contribute to the broader effort of leveraging structured observations to improve the sample efficiency of standard BO techniques (Astudillo & Frazier, 2021b).

### 2.4 CAUSAL BAYESIAN OPTIMIZATION

CBO performs sequential actions to interact with an SCM $\mathcal{M}$. The causal graph structure $\mathcal{G}$ is assumed known, while the functional mechanisms $\mathbf{F} = \{f_i\}_{i=0}^{d}$ are fixed but unknown. CBO uses probabilistic surrogate models - typically Gaussian Processes (GPs Williams & Rasmussen (2006)) - to guide the selection of interventions for maximizing the objective.

In (Aglietti et al., 2020), a CBO algorithm was introduced to jointly identify the optimal intervention set and the corresponding input values that maximize the target variable in an SCM. Dynamic CBO (DCBO) (Aglietti et al., 2021) extends this approach to time-varying SCMs where causal effects evolve over time.

The MCBO method (Sussex et al., 2023) optimizes soft interventions to maximize the target variable within an SCM. In this setting, each edge function becomes $f_i : \mathcal{Z}_i \times \mathcal{A}_i \rightarrow \mathcal{X}_i$. Let $x_{i,t}$ denote the observation of node $X_i$ at time step $t$, for $i \in [d]$ and $t \in [T]$, where $T$ is the total number of time steps. At each step $t$, actions $\mathbf{a}_{:t} = \{\mathbf{a}_{i,t}\}_{i=0}^{d}$ are selected, and the resulting observations $\mathbf{x}_{:,t} = \{x_{i,t}\}_{i=0}^{d}$ are recorded. The relationship between action $\mathbf{a}_{i,t}$ and the observation is modeled using an additive noise structure: $x_{i,t} = f_i(\mathbf{z}_{i,t}, \mathbf{a}_{i,t}) + u_{i,t}, \quad \forall i \in [d]$. For the target node $d$, the action is fixed at $\mathbf{a}_{d,t} = 0$, and the observed outcome is $y_t = f_d(\mathbf{z}_{d,t}, \mathbf{a}_{d,t}) + u_{d,t}$, where $y_t$ depends on the entire intervention vector. The optimal action vector $\mathbf{a}^*$ that maximizes the expected reward is obtained by solving $\mathbf{a}^* = \arg\max_{\mathbf{a} \in \mathcal{A}} \mathbb{E}[y \mid \mathbf{a}]$. A GP surrogate model is employed to approximate the reward function and guide the BO process toward optimizing $y$.

## 3 PROBLEM STATEMENT

Following prior CBO approaches (Aglietti et al., 2020; 2021; Sussex et al., 2023; Frazier, 2018), we assume that the DAG $\mathcal{G}$ is known. Our framework employs GP surrogate models to guide the optimization of soft interventions, which are controlled via an action vector $\mathbf{a} = \{\mathbf{a}_i\}_{i=0}^{d}$, with the goal of maximizing the reward. This section details the specific problem setting addressed in this work.

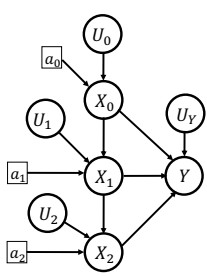

Figure 1: **EXCBO**: Causal Bayesian Optimization via exogenous distribution learning. The distribution of $U_i$ is approximated using the density of the recovered surrogate $\widehat{U}_i$. EXCBO searches for the action vector $\mathbf{a}$ that maximizes the reward $Y$.

### 3.1 ASSUMPTIONS FOR EXCBO

We assume that the causal structure, represented by the DAG $\mathcal{G}$ of the SCM $\mathcal{M} = (\mathcal{G}, \mathbf{F}, \mathbf{V}, \mathbf{U})$, is given. This paper focuses exclusively on this setting. Additionally, we assume that $\mathcal{M}$ is causally sufficient, meaning all endogenous variables in $\mathbf{V}$ are observable. The problems of causal structure learning and handling unobserved confounders are left for future work.

### 3.2 CBO VIA EXOGENOUS DISTRIBUTION LEARNING

In contrast to prior CBO approaches based on ANMs (Aglietti et al., 2021; Sussex et al., 2023), we propose a more flexible modeling of the mappings $f_i()$ by explicitly incorporating exogenous variables. To this end, we introduce EXCBO - a framework for CBO that leverages exogenous distribution learning, as illustrated in Figure 1.

Let $\mathcal{R}$ denote the set of root nodes. Since root nodes have no parents, we set $\mathbf{z}_{i,t} = \mathbf{0}$ for all $i \in \mathcal{R}$. Similarly, we define $\mathbf{a}_{d,t} = 0$ at the target node $d$, and denote the reward at time $t$ as

$y_t = f_d(\mathbf{z}_{d,t}, \mathbf{a}_{d,t}, u_{d,t})$. Given an action vector $\mathbf{a} = \{\mathbf{a}_i\}_{i=0}^d$ and exogenous variables $\mathbf{u} = \{u_i\}_{i=0}^d$, the reward is denoted as $y = \mathbf{F}(\mathbf{a}, \mathbf{u})$. The optimization objective becomes

$$\mathbf{a}^* = \arg\max_{\mathbf{a} \in \mathcal{A}} \mathbb{E}[y \mid \mathbf{a}], \tag{3}$$

where the expectation is taken over the exogenous variables $\mathbf{u}$. The goal is to identify a sequence of interventions $\{\mathbf{a}_t\}_{t=0}^T$ that achieves high average expected reward. To evaluate convergence, we study the cumulative regret over a time horizon $T$: $R_T = \sum_{t=1}^T \left[ \mathbb{E}[y \mid \mathbf{a}^*] - \mathbb{E}[y \mid \mathbf{a}_{:,t}] \right]$. In our experiments, we use the observed objective or reward value $y$ as the primary performance metric for comparing EXCBO against baseline methods. The best choice of evaluation metric may vary depending on the application and the effectiveness of the optimized action sequence.

### 3.3 MOTIVATIONS FOR EXOGENOUS DISTRIBUTION LEARNING

In existing CBO frameworks, the distributions of exogenous variables are either ignored or marginalized to simplify the intervention process (Aglietti et al., 2020; 2021; Sussex et al., 2023). Learning the exogenous distribution, however, yields a more accurate surrogate model when observational data is available. As outlined in later sections, we propose an encoder-decoder architecture (illustrated in Figure 2) to recover the exogenous variable associated with each endogenous node in an SCM. The distribution of an exogenous variable $U_i$ is approximated by the density of its recovered surrogate $\widehat{U}_i$, modeled using a flexible distribution such as a Gaussian Mixture. This learned exogenous distribution improves the surrogate model's approximation of the underlying SCM.

As a result, EXCBO extends beyond the ANM framework assumed by prior work (Aglietti et al., 2020; 2021; Sussex et al., 2023), enabling optimization under a broader class of causal models. Moreover, by enhancing the surrogate model's fidelity, our approach can potentially achieve superior reward outcomes. Additional justification and motivation are provided in the Appendix.

### 3.4 DECOMPOSABLE GENERATION MECHANISM

In our setting, the edges in the SCM $\mathcal{M}$ correspond to a fixed but unknown set of functions $\mathbf{F} = \{f_i\}_{i=0}^d$. We assume the structure of the SCM is known and that the system is causally sufficient—that is, it contains no hidden variables or confounders. We now define the *Decomposable Generation Mechanism (DGM)* used in our analysis.

**Definition 1.** (DGM) A data-generating function $f$ follows a decomposable generation mechanism if $X = f(\mathbf{Z}, U) = f_a(\mathbf{Z}) + f_b(\mathbf{Z}) f_c(U)$, where $f_a : \mathcal{Z} \to \mathbb{R}$, $f_b : \mathcal{Z} \to \mathbb{R}$, and $f_c : \mathcal{U} \to \mathbb{R}$. All mappings are continuous, and $f_b(\mathbf{z}) \neq 0$ for all $\mathbf{z} \in \mathcal{Z}$.

In a DGM, the function $f_c(U)$ may be a one-dimensional, nonlinear, and nonmonotonic transformation of the exogenous variable $U$. The term $f_b(\mathbf{Z}) f_c(U)$ implies that the variance of the generated variable $X$, conditioned on its parents $\mathbf{Z}$, depends on both $U$ and $\mathbf{Z}$. Consequently, DGMs represent a broad class of mechanisms in which both parents and exogenous variables contribute to variance modulation.

This modeling framework is notably more general than *Location-Scale or Heteroscedastic Noise Models (LSNMs)* (Immer et al., 2023), which typically assume linear $f_c()$ and strictly positive $f_b()$. Therefore, DGMs constitute a superset of LSNMs. In Section 4.1, we demonstrate that the distribution of exogenous variables can be recovered when the data-generating mechanism $f$ in each node equation 1 adheres to the DGM formulation.

## 4 EXOGENOUS DISTRIBUTION LEARNING

Given observations of an endogenous node and its parents within an SCM, our goal is to recover the distribution of that node's exogenous variable. This exogenous distribution learning is carried out using GPs. We begin by focusing on the recovery of the exogenous distribution for a single node.

## 4.1 EXOGENOUS VARIABLE RECOVERY FOR ONE NODE

According to equation 2, an endogenous variable $X_i$ may or may not have associated action variables $\mathbf{A}_i$. To simplify notation, we use $\mathbf{Z}_i$ in this section to denote both the parents of $X_i$ and its action variable, i.e., $\mathbf{Z}_i = (\mathbf{Z}_i, \mathbf{A}_i)$ if $X_i \in \mathbf{I}$. The task of learning the exogenous distribution for $X$ then becomes the problem of recovering the distribution of $U$ given observations of $X$ and $\mathbf{Z}$ from the generative model $X = f(\mathbf{Z}, U)$. For clarity, we define the causal mechanism for the triplet $(\mathbf{Z}, U, X)$ corresponding to a single node in an SCM.

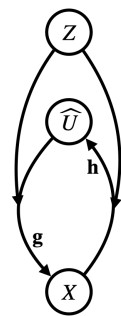

**Assumption 1.** Let $X$ be a node of an SCM $\mathcal{M}$, and let $f()$ be the causal mechanism generating $X$ with parent $\mathbf{Z}$ and an exogenous variable $U$, i.e. $X = f(\mathbf{Z}, U)$. We use $(\mathbf{Z}, U, X, f)$ to denote the node SCM of $X$, and we assume $\mathbf{Z} \perp\!\!\!\perp U$.

We have Assumption 1 for any SCM discussed in this paper. In a node SCM, $\mathbf{Z}$ may be multi-dimensional, representing the parents of $X$, while $U$ is the exogenous variable. This differs from the *Bijective Generation Mechanism* (BGM Nasr-Esfahany et al. (2023)), where $f(\mathbf{Z}, U)$ is assumed to be monotonic and invertible with respect to $U$ given fixed $\mathbf{Z}$.

We adopt an encoder-decoder framework (Figure 2) to construct a surrogate for the exogenous variable. For an observation $(\mathbf{z}, x)$, we use $x(\mathbf{z}, u)$ to denote $f(\mathbf{z}, u)$, and here $u$ is an exogenous value generating $x$. Here $u \sim p(U)$, and $p(U)$ is the exogenous distribution regarding node $X$. The encoder and decoder are learned via BO (Balandat et al., 2020) and a training set that involves $N$ observations or

Figure 2: Structure in one node. $\mathbf{Z}$ denotes the parent set of $X$. Our algorithm learns an encoder $h$ and a decoder $g$ such that the surrogate $\widehat{U} = h(\mathbf{Z}, X)$ and $X = g(\mathbf{Z}, \widehat{U})$.

base samples. By following the analysis of Balandat et al. (2020), we have the following definition.

**Definition 2.** (Encoder-Decoder Surrogate; EDS) Let $(\mathbf{Z}, U, X, f)$ be a node SCM. Let $\phi() : \mathcal{Z} \to \mathcal{X}$ be a probabilistic regression model. Each $\mathbf{z} \in \mathcal{Z}$ has $N$ base samples in the close neighborhood of $\mathbf{z}$, $\left\{\mathbf{z}, x_i(\mathbf{z}, u_i)\right\}_{i=1}^{N}$, and here $\{u_i\}_{i=1}^{N}$ $i.i.d \sim p(U)$. In addition, $\phi()$ has a prediction mean $\mu_\phi(\mathbf{z}) = \frac{1}{N} \sum_i^N x_i(\mathbf{z}, u_i)$ and a variance $\sigma_\phi^2(\mathbf{z}) = \frac{1}{N-1} \sum_i^N \left(x_i(\mathbf{z}, u_i) - \mu_\phi(\mathbf{z})\right)^2$. We define $(\widehat{U}, \phi, h, g)$ as an encoder-decoder surrogate (EDS) for the exogenous variable $U$, where the encoder is $h() : \mathcal{Z} \times \mathcal{X} \to \widehat{\mathcal{U}}$, defined as $\widehat{U} := h(\mathbf{Z}, X) := \frac{X - \mu_\phi(\mathbf{Z})}{\sigma_\phi(\mathbf{Z})}$, and the decoder is $g() : \mathcal{Z} \times \widehat{\mathcal{U}} \to \mathcal{X}$.

Given observations of $X$ and its parents $\mathbf{Z}$, our method learns the encoder $h()$ to approximate the true value of $U$ via $\widehat{u} = h(\mathbf{z}, x)$. Concurrently, the decoder $g()$ serves as a surrogate for the causal mechanism $f()$, reconstructing $x = g(\mathbf{z}, \widehat{u})$. Theorem 4.1 establishes that surrogate values of the exogenous variable $U$ can be recovered from observations under the DGM assumption on $f$.

**Theorem 4.1.** *Let $(\mathbf{Z}, U, X, f)$ be a node SCM, and $(\widehat{U}, \phi, h, g)$ an EDS surrogate of $U$. Suppose $f$ has the DGM structure, i.e. $X = f(\mathbf{Z}, U) = f_a(\mathbf{Z}) + f_b(\mathbf{Z})f_c(U)$ with $f_b(\mathbf{z}) \neq 0$ for all $\mathbf{z} \in \mathcal{Z}$. In addition, each $\mathbf{z} \in \mathcal{Z}$ has $N$ base samples in the close neighborhood of $\mathbf{z}$, i.e., $\left\{\mathbf{z}, x_i(\mathbf{z}, u_i)\right\}_{i=1}^{N}$ with $\{u_i\}_{i=1}^{N}$ $i.i.d \sim p(U)$. Then with $N \to \infty$, the surrogate $\widehat{U} \to \frac{s}{\sigma_{f_c}}\left(f_c(U) - \mathbb{E}[f_c(U)]\right)$, $\mathbb{E}[\widehat{U}] \to 0$, $\mathrm{Var}[\widehat{U}] \to 1$, and $\widehat{U} \perp\!\!\!\perp \mathbf{Z}$, where $\sigma_{f_c} = \sqrt{\mathbb{E}\left[\left(f_c(U) - \mathbb{E}[f_c(U)]\right)^2\right]}$, $s \in \{-1, 1\}$.*

We use the distribution of the recovered surrogate $\widehat{U} = \mathbf{s}(U) = h(\mathbf{Z}, X)$ - denoted as $p(\widehat{U})$ - as a proxy for the true $p(U)$ in the surrogate model. Consequently, the function $f$ is approximated via the learned decoder $g$ and the surrogate $\widehat{u}$:

$$x = f(\mathbf{z}, u) = g(\mathbf{z}, \widehat{u}) = g(\mathbf{z}, \mathbf{s}(u)).$$

Figure 3 illustrates the relationship among different data generation mechanisms regarding counterfactual identifiability. Definition and analysis on counterfactual identifiability can be found

in Appendix-F. Notably, our framework generalizes beyond ANM (linear), and BGM (monotonic) to a new class of nonlinear and nonmonotonic models through DGM. This extends the identifiability of $U$ significantly beyond the standard assumption $X = f(\mathbf{pa}(X)) + U$ used in many BO and CBO methods. We use EDS$^*$ to represent the node SCMs that are counterfactually identifiable via EDS either with or without the condition of $\widehat{U} \perp\!\!\!\perp \mathbf{Z}$.

The proof of Theorem 4.1 is provided in Appendix E. Our surrogate variable $\widehat{U}$ and encoder $h()$ are valid under both DGM and BGM (Nasr-Esfahany et al., 2023) assumptions. In the BGM case, recovery of $U$ requires enforcing $\widehat{U} \perp\!\!\!\perp \mathbf{Z}$, as detailed in Appendix F, which can be achieved through independence regularization - albeit at additional computational cost. If $f$ does not satisfy the DGM or BGM assumptions, then the recovered $\widehat{U}$ may be dependent on $\mathbf{Z}$, potentially degrading the accuracy of the surrogate model and limiting the effectiveness of CBO in finding optimal $y$ using limited data.

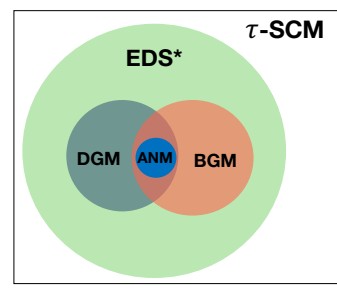

Figure 3: Scopes of different mechanism classes.

## 4.2 IMPLEMENTATION OF EXOGENOUS DISTRIBUTION LEARNING

The encoder-decoder architecture in Figure 2 can be implemented in various ways, such as using Variational Autoencoders (VAEs) (Kingma & Welling, 2014) or sample efficient deep-generative models (Liang et al., 2024; Wang et al., 2023). To keep the implementation straightforward, we adopt GP regression for both the encoder and decoder, consistent with the EDS definition in Definition 2.

For nodes with action variables $\mathbf{A}$, the decoder becomes $g() : \mathcal{Z} \times \mathcal{A} \times \widehat{\mathcal{U}} \to \mathcal{X}$, and the encoder becomes $h() : \mathcal{Z} \times \mathcal{A} \times \mathcal{X} \to \widehat{\mathcal{U}}$, while the regression model is $\phi() : \mathcal{Z} \times \mathcal{A} \to \mathcal{X}$. Both $g()$ and $\phi()$ are implemented using GP regression models (Williams & Rasmussen, 2006). To approximate the distribution of the recovered exogenous surrogate $\widehat{U}$, we use a Gaussian Mixture model to estimate $p(\widehat{U})$, which serves as a replacement for $p(U)$ in the probabilistic surrogate objective. For all nodes in the SCM $\mathcal{M}$, we denote the collection of decoders as $\mathbf{G} = \{g_i\}_{i=0}^d$ and the collection of encoders as $\mathbf{H} = \{h_i\}_{i=0}^d$.

## 5 CBO WITH EXOGENOUS DISTRIBUTION LEARNING

In this section, we present the EXCBO algorithm, describing the probabilistic model and acquisition function used.

## 5.1 STATISTICAL MODEL

In our model, the function $f_i$ that generates variable $X_i$ is learned through $g_i$, and $X_i = g_i(\mathbf{Z}_i, \mathbf{A}_i, \widehat{U}_i)$. We use GPs (Williams & Rasmussen, 2006) to learn the surrogate of $g_i$, i.e., $\tilde{g}_i$. For $i \in [d]$, let $\mu_{g,i,0}$ and $\sigma_{g,i,0}$ denote the prior mean and variance function for each $f_i$, respectively. At step $t$, the observation set is $\mathcal{D}_t = \{\mathbf{z}_{:,1:t}, \mathbf{a}_{:,1:t}, x_{:,1:t}\}$. The posterior of $g_i$ with the input of node $i$, $(\mathbf{z}_i, \mathbf{a}_i, \widehat{u}_i)$, is given by

$$g_{i,t}(\mathbf{z}_i, \mathbf{a}_i, \widehat{u}_i) \sim \mathcal{GP}(\mu_{g,i,t-1}, \sigma_{g,i,t-1}^2); \; \mu_{g,i,t-1} = \mu_{g,i,t-1}(\mathbf{z}_i, \mathbf{a}_i, \widehat{u}_i); \sigma_{g,i,t-1} = \sigma_{g,i,t-1}(\mathbf{z}_i, \mathbf{a}_i, \widehat{u}_i).$$

Then $x_{i,t} = g_{i,t}(\mathbf{z}_i, \mathbf{a}_i, \widehat{u}_i)$ denotes observations from one of the plausible models. Here $\widehat{u}_i \sim p(\widehat{U}_i)$ in the sampling of the learned distribution of $\widehat{U}_i$.

Given an observation $(\mathbf{z}_i, \mathbf{a}_i, x_i)$ at node $i$, the exogenous recovery $\widehat{u}_i = h_i(\mathbf{z}_i, \mathbf{a}_i, x_i) = \frac{x_i - \mu_{\phi,i}(\mathbf{z}_i, \mathbf{a}_i)}{\sigma_{\phi,i}(\mathbf{z}_i, \mathbf{a}_i)}$. At time step $t$, the posterior of $\phi_i$ with the input of node $i$, $(\mathbf{z}_i, \mathbf{a}_i)$, is given by

$$\phi_{i,t}(\mathbf{z}_i, \mathbf{a}_i) \sim \mathcal{GP}(\mu_{\phi,i,t-1}(\mathbf{z}_i, \mathbf{a}_i), \sigma_{\phi,i,t-1}^2(\mathbf{z}_i, \mathbf{a}_i)) \tag{4}$$

Therefore, $\widehat{u}_i = h_{i,t}(\mathbf{z}_i, \mathbf{a}_i, x_i) = \frac{x_i - \mu_{\phi,i,t-1}(\mathbf{z}_i, \mathbf{a}_i)}{\sigma_{\phi,i,t-1}(\mathbf{z}_i, \mathbf{a}_i)}$. According to the definition of $h()$ in Theorem 4.1, $h()$ also follows a GP, i.e. $h_{i,t}(\mathbf{z}_i, \mathbf{a}_i, x_i) \sim \mathcal{GP}(\mu_{h,i,t-1}, \sigma_{h,i,t-1}^2)$. This GP is defined

by $\phi_{i,t}()$ which is sampled with equation 4. Different from $g_i()$, the observations of the input $(\mathbf{Z}_i, \mathbf{A}_i, X_i)$ for $h_i()$ are only required at the training time, and we only need to sample the learned $p(\widehat{U}_i)$ to get value $\widehat{u}_i$ for model prediction or model sampling.

## 5.2 ACQUISITION FUNCTION

Algorithm 1 describes the proposed method solving equation 3. In iteration $t$, it uses GP posterior belief of $y$ to construct an upper confidence bound (UCB Brochu et al. (2010); Frazier (2018)) of $y$:

$$\mathrm{UCB}_{t-1}(\mathbf{a}) = \mu_{t-1}(\mathbf{a}) + \beta_t \sigma_{t-1}(\mathbf{a}). \tag{5}$$

Here $\mu_{t-1}(\mathbf{a}) = \mathbb{E}[\mu_{g,d,t-1}(\mathbf{z}_d, \mathbf{a}_d, \widehat{u}_d)]$ ; $\sigma_{t-1}(\mathbf{a}) = \mathbb{E}[\sigma_{g,d,t-1}(\mathbf{z}_d, \mathbf{a}_d, \widehat{u}_d)]$, where the expectation is taken over $p(\widehat{U})$. In equation 5, $\beta_t$ controls the tradeoff between exploration and exploitation of Algorithm 1. The UCB-based algorithm is a classic strategy that is widely used in BO and stochastic bandits (Lattimore & Szepesvári, 2020; Srinivas et al., 2010). The proposed algorithm adapts the "optimism in the face of uncertainty" (OFU) strategy by taking the expectation of the UCB as part of the acquisition process.

## 5.3 ALGORITHM

Let $k_{g,i}, k_{\phi,i}, \forall i \in [d]$ represent the kernel functions of $g_i$ and $\phi_i$. The proposed EXCBO algorithm is summarized by Algorithm 1. In each iteration, a new sample is observed according to the UCB values. Then the posteriors of $\mathbf{G}$ and $\mathbf{H}$ are updated with the new dataset. The next section gives a theoretical analysis of the algorithm.

---

**Algorithm 1** EXCBO

**Input:** $k_{g,i}, k_{\phi,i}, \forall i \in [d]$
**Result:** Intervention actions $\mathbf{a}_i, \forall i \in [d]$
**for** $t = 1$ **to** $T$ **do**
  Find $\mathbf{a}_t$ by optimizing the acquisition function, $\mathbf{a}_t \in \arg\max \mathrm{UCB}_{t-1}(\mathbf{a})$;
  Observe samples $\{\mathbf{z}_{i,t}, x_{i,t}\}_{i=0}^d$ with the action sequence $\mathbf{a}_t$ and update $\mathcal{D}_t$;
  Use $\mathcal{D}_t$ to update posteriors $\{\mu_{\phi,i,t}, \sigma_{\phi,i,t}^2\}_{i=0}^d$ and exogenous surrogate $\{\widehat{u}_{i,t}\}_{i=0}^d$;
  Use $\mathcal{D}_t \cup \{\widehat{u}_{i,t}\}_{i=0}^d$ to update the decoder posteriors $\{\mu_{g,i,t}, \sigma_{g,i,t}^2\}_{i=0}^d$ ;
**end for**

---

## 6 REGRET ANALYSIS

This section describes the convergence guarantees for EXCBO using soft interventions. Our analysis shows that EXCBO has a sublinear cumulative regret bound (Sussex et al., 2023). In DAG $\mathcal{G}$ over $\{X_i\}_{i=0}^d$, let $N$ be the maximum distance from a root to $X_d$, i.e., $N = \max_i \mathrm{dist}(X_i, X_d)$. Here $\mathrm{dist}(\cdot, \cdot)$ is a measure of the edges in the longest path from $X_i$ to the reward node $Y := X_d$. Let $M$ denote the maximum number of parents of any variables in $\mathcal{G}$, $M = \max_i |\mathbf{pa}(i)|$. Let $L_t$ be a function of $L_g$, $L_{\sigma_g}$, and $N$. With Assumptions 2- 4 in the Appendix, the following theorem bounds the performance of EXCBO in terms of cumulative regret. We present the assumptions used in the regret analysis in Appendix G. Assumption 2 gives the Lipschitz conditions of $g_i$, $\sigma_{g,i}$, and $\mu_{g,i}$. It holds if the RKHS of each $g_i$ has a Lipschitz continuous kernel (Curi et al., 2020; Sussex et al., 2023). Assumption 4 holds when we assume that the $i$th GP prior uses the same kernel as the RKHS of $g_i$ and that $\beta_{i,t}$ is sufficiently large to ensure the confidence bounds in

$$\left| g_i(\mathbf{z}_i, \mathbf{a}_i, \widehat{u}_i) - \mu_{g,i,t-1}(\mathbf{z}_i, \mathbf{a}_i, \widehat{u}_i) \right| \le \beta_{i,t} \sigma_{g,i,t-1}(\mathbf{z}_i, \mathbf{a}_i, \widehat{u}_i) , \quad \forall \mathbf{z}_i \in \mathcal{Z}_i, \mathbf{a}_i \in \mathcal{A}_i, \widehat{u}_i \in \widehat{\mathcal{U}}_i.$$

**Theorem 6.1.** *Consider the optimization problem in equation 3, with the SCM satisfying Assumptions 2- 4, where $\mathcal{G}$ is known but $\mathbf{F}$ is unknown. Then with probability at least $1 - \alpha$, the cumulative regret of Algorithm 1 is bounded by $R_T \le \mathcal{O}(L_T M^N d \sqrt{T \gamma_T})$.*

Here $\gamma_T = \max_t \gamma_{i,T}$ denote the maximum information gain at time $T$. The proof of Theorem 6.1 and further analysis can be found in Appendix G.

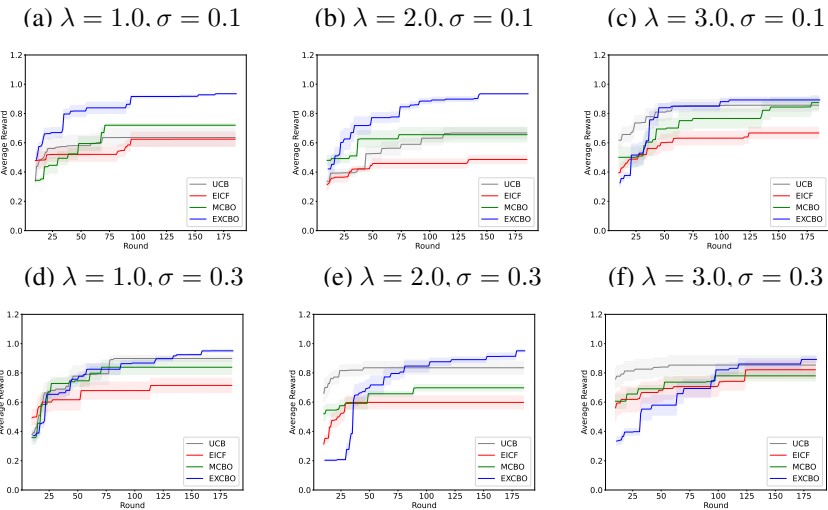

Figure 4: Results of Dropwave with $\sigma \in \{0.1, 0.3\}$ and $\lambda \in \{1.0, 2.0, 3.0\}$.

## 7 EXPERIMENTAL STUDY

This section presents experimental comparisons of the proposed EXCBO and existing algorithms. Different from the single-mode Gaussian noise in MCBO (Sussex et al., 2023), We use two-mode exogenous distributions in the synthetic datasets, i.e.

$$p(U) = w_1 \mathcal{N}(\mu_1, c_1\sigma^2) + w_2 \mathcal{N}(\mu_2, c_2\sigma^2), \quad w_1, w_2, c_1, c_2 > 0, w_1 + w_2 = 1.0. \tag{6}$$

Additional experimental results and analysis are presented in Appendix D.

### 7.1 BASELINES

We compare EXCBO against three representative algorithms: UCB (Brochu et al., 2010; Frazier, 2018), EICF (Astudillo & Frazier, 2019), and MCBO (Sussex et al., 2023). UCB is a standard Bayesian Optimization (BO) method (Brochu et al., 2010; Frazier, 2018), EICF applies a composite function approach to BO, and MCBO is a Causal Bayesian Optimization method discussed in previous sections. Unlike the other baselines, MCBO incorporates neural networks alongside GPs to capture model uncertainty. All algorithms are implemented in Python using the BoTorch library (Balandat et al., 2020).

For MCBO, we adopt the default initial observation size recommended in its original work, which is $2(|\mathbf{A}| + 1)$, where $|\mathbf{A}|$ denotes the number of action variables. For the other methods, the initial sample size ranges from 5 to 20. Each algorithm is executed four times with different random seeds to compute the mean and standard deviation of the resulting reward values.

### 7.2 DROPWAVE

There are two endogenous nodes in Dropwave, i.e., $X$ and the target node $Y$ (Figure 8 in D.2). There are two action nodes associated with $X$, i.e. $a_0, a_1 \in [0, 1]$. Here $X = \sqrt{(10.24a_0 - 5.12)^2 + (10.24a_1 - 5.12)^2} + \lambda U_X$, and $Y = (1.0 + \cos(12.0X))/(2.0 + 0.5X^2) + \lambda U_Y$, $U_X \sim p(U_X)$, and $U_Y \sim p(U_Y)$. We vary $\sigma$ and $\lambda$ to simulate different levels of noise. While $\sigma$ controls the variance of the exogenous variables ($U_X$ and $U_Y$), $\lambda$ scales their effect on the target variable $Y$. Figure 4 presents performance results under various $\sigma$ and $\lambda$ settings. In this set of exeriments, EXCBO outperforms UCB and EICF in most cases, except when $\sigma = 0.1$ and $\lambda = 1.0$.

### 7.3 ALPINE2

We study the algorithms using the Alpine2 dataset (Sussex et al., 2023). There are six endogenous nodes in the Alpine2 dataset as shown in Figure10. In first set of experiments, Alpine2 is generated via DGM with multimodal exogenous distributions as given in equation 7 in Section D.3.1. The results of Alpine2 are shown in Figures 5. We also compared the algorithms on Non-DGM generated

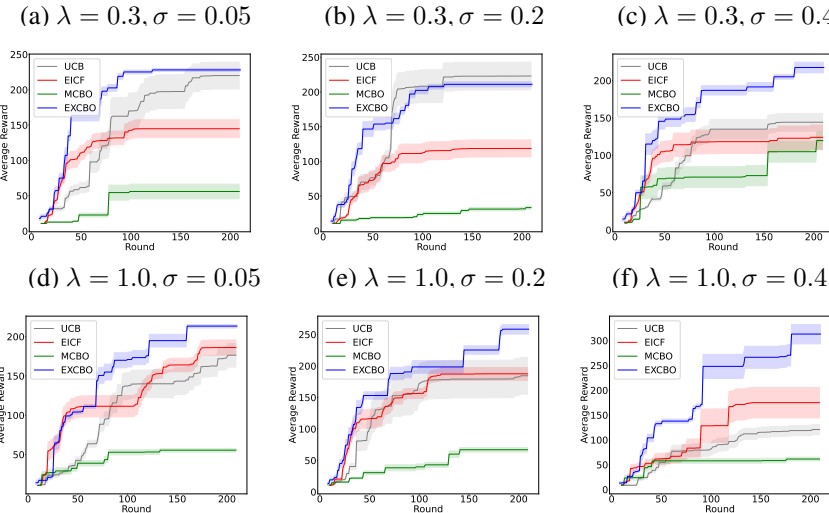

(a) $\lambda = 0.3, \sigma = 0.05$    (b) $\lambda = 0.3, \sigma = 0.2$    (c) $\lambda = 0.3, \sigma = 0.4$

(d) $\lambda = 1.0, \sigma = 0.05$    (e) $\lambda = 1.0, \sigma = 0.2$    (f) $\lambda = 1.0, \sigma = 0.4$

Figure 5: Results of Alpine2 with $\sigma \in \{0.05, 0.2, 0.4\}$ and $\lambda \in \{0.3, 1.0\}$.

Alpine2 dataset in Section D.3.2. As shown in the plots, our EXCBO outperforms the other methods at different noise levels. It shows the effectiveness and benefits of the proposed EXCBO method in multimodal exogenous distribution and mechanism learning.

### 7.4 EPIDEMIC MODEL CALIBRATION

We test EXCBO on an epidemic model calibration by following the setup in Astudillo & Frazier (2021a). In this model, as shown in Figure 6-(c), $I_{i,t}$ represents the fraction of the population in group $i$ that are "infectious" at time $t$; $\beta_{i,j,t}$ is the rate of the people from group $i$ who are "susceptible" have close physical contact with people in group $j$ who are "infectious" at time $t$. We assume there are two groups, and infections resolve at a rate of $\gamma$ per period. The number of infectious individuals in group $i$ at the start of the next time period is $I_{i,t+1} = I_{i,t}(1 - \gamma) + (1 - I_{i,t}) \sum_j \beta_{i,j,t} I_{j,t}$. We assume each $I_{i,t}$ has an observation noise $U_{i,t}$. The model calibration problem is that given limited noisy observations of $I_{i,t}$s, how to efficiently find the $\beta_{i,j,t}$ values in the model. The reward is defined as the negative mean square error (MSE) of all the $I_{i,t}$ observations as the objective function to optimize. In this model, $\beta_{i,j,t}$s are the action variables. The noise is added with two-mode as in equation 6 under ANM (Hoyer et al., 2008). Figure 6-(a-b) visualize the results at the noise levels with $\sigma = 0.1$ and $\sigma = 0.3$.

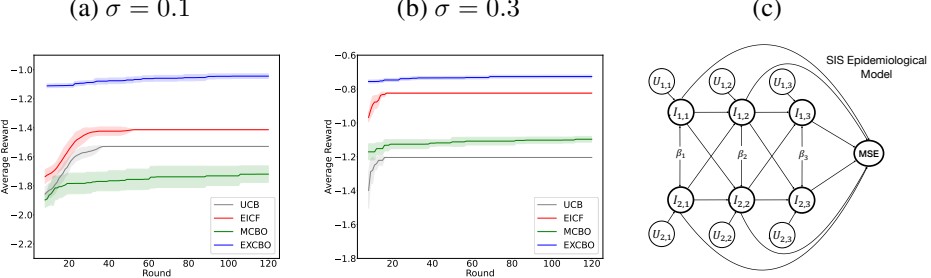

(a) $\sigma = 0.1$     (b) $\sigma = 0.3$     (c)

Figure 6: (a-b): Results of epidemic model calibration; (c): Graph structure for epidemic model calibration.

### 7.5 PLANKTONIC PREDATOR–PREY COMMUNITY IN A CHEMOSTAT

We evaluate the algorithms on a *real-world* dataset from the **p**lanktonic **p**redator–**p**rey **c**ommunity in a **c**hemostat (P3C$^2$). This biological system involves two interacting species, one predator and one prey, and our objective is to identify interventions that reduce the concentration of dead animals in the chemostat, $D_t$. We adopt the system of ordinary differential equations (ODE) from Blasius et al. (2020); Aglietti et al. (2021) as the SCM, and construct the DAG by unrolling the temporal dependencies of two adjacent time steps. Observational data from Blasius et al. (2020) are used

to compute the dynamic causal prior. Unlike dynamic sequential CBO (Aglietti et al., 2021), we employ the causal structure at $t$ and $t+1$ as the DAG for the algorithms. Figure 7 compares the performance of EXCBO with baselines. Additional experimental detai

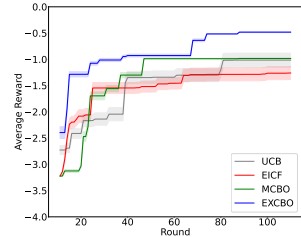

## 8 CONCLUSIONS

We propose a novel CBO algorithm, EXCBO, that approximately recovers the exogenous variables in a structured causal model. With the recovered exogenous distribution, our method naturally improves the surrogate model's accuracy in the approximation of the SCM. Furthermore, the recovered exogenous variables may enhance the surrogate model's capability in causal inference and hence improve the reward values attained by EXCBO. We additionally provide theoretical analysis on both exogenous variable recovery and the algorithm's cumulative regret bound. Experiments on multiple datasets show the algorithm's soundness and benefits.

Figure 7: Results of P3C$^2$ dataset; the reward $y = -D_t$.

## ETHICS STATEMENT

This study relies exclusively on synthetic and publicly available datasets, without the involvement of human subjects or sensitive personal data. Therefore, we do not anticipate any ethical concerns related to this work.

## REPRODUCIBILITY STATEMENT

The assumptions and definitions of DGM are presented in Section 3.4. The assumptions and theoretical foundations for exogenous distribution learning are provided in Section 4.1 and in Sections E and F of the Appendix. The implementation details of EXCBO are discussed in Sections 4.2 and 5. Experimental details, including the generation of synthetic data, processing of real-world data, and overall experimental setups, are presented in Section 7 and in Section D of the Appendix. The assumptions and proof steps for the regret analysis are given in Section G of the Appendix. The code and all datasets used in the paper will be released publicly after the review period.

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

## A STATMENT ON THE USE OF LARGE LANGUAGE MODELS (LLMS)

The authors acknowledge the use of large language models (LLMs) for the limited purpose of grammar checking and language polishing. No LLMs were used for data analysis, methodological design, or generation of scientific content. All ideas, results, and conclusions presented in this manuscript are the full responsibility of the authors.

## B ADDITIONAL REMARKS

### B.1 REMARKS ON MOTIVATIONS

Learning the exogenous distribution enhances the surrogate model's ability to approximate the ground truth SCMs. As discussed in Sections 4.1,E, andF, under moderate assumptions, the independence between the recovered exogenous variable $\widehat{U}$ and both the parents $\mathbf{Z}$ and actions $\mathbf{A}$ empower the structured surrogate model in EXCBO to perform effective intervention inference. This independence reduces the influence of environmental noise or exogenous variables on the actions or interventions derived from the acquisition function.

This work considers the setting where the causal structure is known, and the model $\mathcal{M}$ is causally sufficient. The challenges of learning causal structures and dealing with unobserved confounders are left for future research.

We believe multi-modal and non-Gaussian exogenous distributions are prevalent in real-world systems. When each exogenous variable is viewed as an unobserved latent factor, it is highly plausible that such factors follow non-Gaussian distributions with multiple modes.

### B.2 PERFORMANCE GAPS

UCB, EICF, and MCBO use $X = \widehat{f}(\mathbf{Z}, \mathbf{A})$ or $X = \widehat{f}(\mathbf{Z}, \mathbf{A}, \epsilon), \epsilon \in \mathcal{N}(0, 1)$ to approximate $X = f(\mathbf{Z}, \mathbf{A}, U)$ for each node or the overall reward function. The absence of information about $U$ introduces irreducible bias into the surrogate model of the reward function. In contrast, EXCBO explicitly recovers the exogenous variable $U$ and learns its multi-modal distribution, producing a more accurate surrogate, i.e., $X = \widehat{f}(\mathbf{Z}, \mathbf{A}, \widehat{U})$, for the objective reward function, even when the variance $\sigma^2$ in the data is small. Experimental results further show that EXCBO enhances the robustness of CBO, particularly in scenarios with limited data samples.

### B.3 BROADER IMPACTS

As a new causal Bayesian optimization framework, EXCBO may help reduce the required training samples for more efficient and cost-effective decision-making, which may have broader impacts in many science and engineering applications, such as future pandemic preparedness with better-calibrated epidemic dynamic models as illustrated in the paper. However, if misused, the societal consequences of designing new systems or materials with unforeseen future threats has to be taken into consideration with caution.

## C Nomenclature

| Symbol | Description |
|---|---|
| $U$ | a single exogenous variable |
| $\mathbf{U}$ | the exogenous variable set of a SCM, i.e., $\mathbf{U} = \{U_1, ..., U_d\}$ |
| $\widehat{U}$ | the exogenous variable recovered via EDS, i.e., the EDS surrogate of $U$ |
| $u$ | a value or realization of variable $U$ |
| $\widehat{u}$ | a value or realization of variable $\widehat{U}$ |
| $\mathcal{U}$ | the domain, or value space of variable $U$ |
| $\widehat{\mathcal{U}}$ | the domain, or value space of variable $\widehat{U}$ |
| $h_i()$ | the EDS encoder function for node $X_i$ |
| $\widehat{u}_i$ | the output of $h_i()$ given an input |
| $\tilde{h}_i()$ | a plausible function of $h_i$ via posterior GP trained with data in some step of EXCBO |
| $\widehat{\tilde{u}}_i$ | the output of $\tilde{h}_i()$ given an input |
| $\mathbf{Z}_i$ | the parent of $X_i$, $\mathbf{pa}(i)$ |
| $g_i()$ | the EDS decoder function for node $X_i$ |
| $\mathbf{G}$ | the collection of decoders, $\mathbf{G} = \{g_i\}_{i=0}^d$ |
| $\mathbf{H}$ | the collection of encoders, $\mathbf{H} = \{h_i\}_{i=0}^d$ |

## D Additional Experimental Results and Analysis

In our experiments, the synthetic data are generated via ANM, DGM, and Non-DGM mechanisms.

### D.1 Experimental Setup

We report the expected reward, $\mathbb{E}_U[y \mid \mathbf{a}_t]$, as a function of the number of system interventions performed. Each figure presents the mean performance over four random seeds, with error bars representing the interval $[-0.2\sigma, 0.2\sigma]$. The GPs used in our models are implemented via the `SingleTaskGP()` function from BoTorch (Balandat et al., 2020), and are trained using the default hyperparameters described in Hvarfner et al. (2024). Each Gaussian Mixture Model (GMM) has two components. Action node domains are normalized to lie within $[0, 1]$. To reduce computational overhead, we restrict the number of $\sigma$ values considered for the exogenous variables in each dataset.

### D.2 Dropwave

In Dropwave Dataset, the values of action nodes $a_0, a_1 \in [0, 1]$, $X = \sqrt{(10.24a_0 - 5.12)^2 + (10.24a_1 - 5.12)^2} + \lambda U_X$, and $Y = (1.0 + \cos(12.0X))/(2.0 + 0.5X^2) + \lambda U_Y$, $U_X \sim p(U_X)$, and $U_Y \sim p(U_Y)$. Here $p(U_X) = 0.5\mathcal{N}(-0.2, , 1.4\sigma^2) + 0.5\mathcal{N}(0.4, \sigma^2)$, and $p(U_Y) = 0.5\mathcal{N}(-0.1, 0.32\sigma^2) + 0.5\mathcal{N}(0.05, 0.32\sigma^2)$. Clearly, the data generation here belongs to the ANMs (Hoyer et al., 2008).

As shown in the plots, UCB's performance improves with increasing $\sigma$ or $\lambda$, suggesting that strong exogenous noise may diminish the benefits of structural knowledge utilized by EICF and EXCBO. Nevertheless, EXCBO still achieves superior (Figure 4-f) or comparable (Figure 4-c) performance even under high-noise conditions.

Figure 9 gives the results of EXCBO on three datasets using different GMM component numbers. We can see that EXCBO models with different GMM component numbers give similar results.

### D.3 Alpine2

The Alpine2 dataset contains six endogenous nodes, as illustrated in Figure 10. The exogenous distributions for $X$ and $Y$ follow Gaussian Mixture models with two components, as defined in equation equation 6. Due to the high computational cost of evaluating MCBO (Sussex et al., 2023),

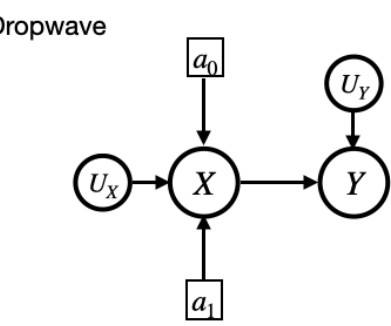

Figure 8: Graph structure of Dropwave dataset.

(a) Dropwave, $\lambda = 1.0, \sigma = 0.1$;  (b) Alpine2, $\lambda = 2.0, \sigma = 0.1$;  (c)  P3C$^2$

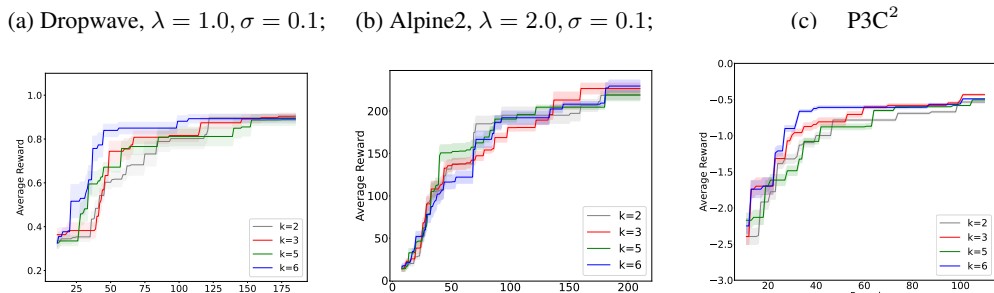

Figure 9: Results of EXCBO using different number of GMM components.

we restrict our comparisons in this experiment to UCB (Brochu et al., 2010; Frazier, 2018) and EICF (Astudillo & Frazier, 2019).

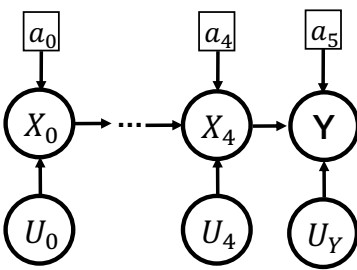

Figure 10: Graph structure of the Alpine2 dataset.

### D.3.1 ALPINE2 WITH DGM MECHANISM

We evaluate the algorithms on the synthetic Alpine2 dataset (Sussex et al., 2023), generated using a DGM mechanism with multimodal exogenous distributions. Each node is defined as

$$X_0 = -\sqrt{10.0a_0} \sin(10.0a_0) + \big(\cos(10.0a_0) + 1.2\big) \cdot \lambda U_0^4; \tag{7}$$
$$X_i = \sqrt{10.0a_i} \sin(10.0a_i) X_{i-1} + 0.1\big(\cos(10.0a_i) + X_{i-1}^2 + 1.2\big) \cdot \lambda U_i^4, \ 1 \le i \le 5;$$
$$Y = X_5.$$

Here, $U_i \sim p(U_i)$ as specified in equation 6, with $w_1 = w_2 = 0.5$, $\mu \in [-1.0, 1.0]$, and $c_1, c_2 \in [0.05, 1.5]$. The results for $\sigma \in \{0.05, 0.2, 0.4\}$ and $\lambda \in \{0.3, 1.0\}$ are shown in Figure 5.

### D.3.2 ALPINE2 WITH NON-DGM MECHANISM

For the non-DGM setting of the Alpine2 dataset (Sussex et al., 2023), each node is defined as

$$X_0 = -\sqrt{10.0a_0 + U_0}\sin(10.0a_0 + U_0); \tag{8}$$
$$X_i = \sqrt{10.0a_i + U_i}\sin(10.0a_i + U_i)X_{i-1},\ 1 \le i \le 5;$$
$$Y = X_5.$$

Here, $U_i \sim p(U_i)$ as defined in equation 6, with $w_1 = w_2 = 0.5$, $\mu \in [-1.0, 1.0]$, and $c_1, c_2 \in [0.05, 1.5]$.

Due to computational constraints, we use $\sigma \in \{0.05, 0.1, 0.2\}$. The corresponding results are reported in Figures 11-(a–c). As shown, EXCBO consistently achieves the best performance across all noise levels, demonstrating the effectiveness and advantages of the proposed method. Although the Alpine2 generation mechanism does not strictly follow DGM or BGM, the strong results of EXCBO, as illustrated in Figures 11-(a–c), highlight its generalization capability, providing further empirical support for the theoretical claims in Sections 4.1 and F.



(a) $\sigma = 0.05$  (b) $\sigma = 0.1$  (c) $\sigma = 0.2$



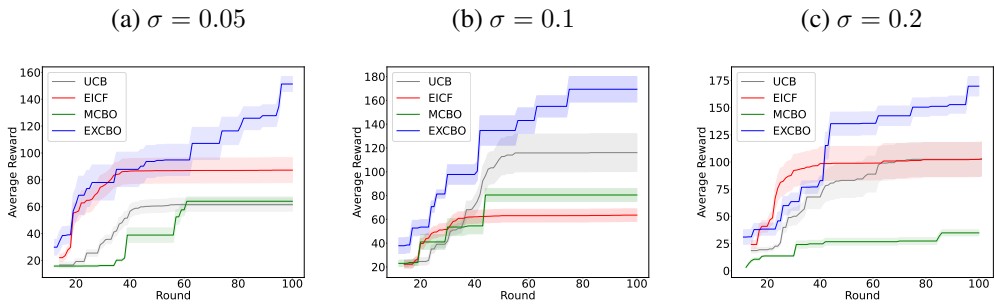

Figure 11: (a-c): Results of Alpine2 (generated via Non-DGM mechanism in equation 8).

### D.4 EPIDEMIC MODEL CALIBRATION

We adopt the additive noise model (ANM Hoyer et al. (2008)), i.e., $X_i = f(\mathbf{Z}_i) + U_i$, where $U_i \sim p(U) = 0.5\mathcal{N}(\mu_1, c_1\sigma^2) + 0.5\mathcal{N}(\mu_2, c_2\sigma^2),\ c_1, c_2 > 0$. Since ANM is a subset of DGM, this setup also satisfies the DGM assumption. To ensure consistency, we normalize and standardize all action nodes to the range $[0, 1]$. Specifically, $\beta$ is rescaled to $[0, 1]$, with $\gamma = 0.5$, $I_{i,0} = 0.01$ for $i \in \{0, 1\}$, and $T = 3$. For $U_{i,j}$ with $i \in \{1, 2\}$ and $j \in \{1, 2, 3\}$, we set $w_1 = w_2 = 0.5$, $\mu_1, \mu_2 \in [-1.0, 1.0]$, and $c_1, c_2 \in \{0.5, 1.0, 1.5\}$. With the capability to recover and learn the exogenous distributions, our method is more robust and stable in this application scenario. Similarly constrained by computational overhead, we use $\sigma \in \{0.1, 0.3\}$, with the other $p(U)$ hyperparameters set as in the Alpine2 experiments. Figure 6 shows that increased exogenous noise enhances the performance of all methods. Our EXCBO performs better than state-of-the-art model calibration methods in both cases, and our method has a faster convergence rate compared to the baselines.

### D.5 PLANKTONIC PREDATOR–PREY COMMUNITY IN A CHEMOSTAT

We use the system of ordinary differential equations (ODE) given by Blasius et al. (2020); Aglietti et al. (2021) as our SCM and construct the DAG by rolling out the temporal variable dependencies in the ODE of two adjacent time steps while removing graph cycles. Observational data are provided in Blasius et al. (2020), and are use to compute the dynamic causal prior. So different from dynamic sequential CBO (Aglietti et al., 2021), we use the causal structure at $t$ and $t+1$ as the DAG for the algorithms. The causal graph is given in Figure 12.

At each time step, the system includes the following variables:

- $N_{in}$: Nitrogen concentration in the external medium

- $N$: Nitrogen (prey) concentration

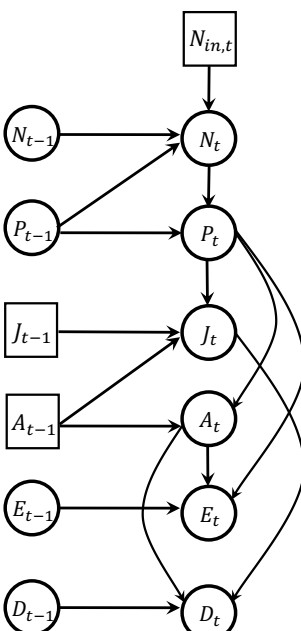

Figure 12: P3C$^2$ graph structure; exogenous nodes are not included.

- $P$: Phytoplankton (predator) concentration

- $E$: Predator egg concentration

- $J$: Predator juvenile concentration

- **A**: Predator adult concentration

- $D$: Dead animal concentration

Equations (21–26) in Aglietti et al. (2021) define the ODE, and equations (9-14) specify the corresponding SCM. The action variables are $N_{in,t}$, $J_t$, and $A_t$, which we manipulate to minimize $D_{t+1}$. We use GPs to fit the following SCM

$$N_t = f_N(N_{in,t}, N_{t-1}, P_{t-1}, \epsilon_N) \tag{9}$$
$$P_t = f_P(N_t, P_{t-1}, \epsilon_P) \tag{10}$$
$$J_t = f_J(P_t, J_{t-1}, A_{t-1}, \epsilon_J) \tag{11}$$
$$A_t = f_A(P_t, A_{t-1}, \epsilon_A) \tag{12}$$
$$E_t = f_E(P_t, A_t, E_{t-1}, \epsilon_E) \tag{13}$$
$$D_t = f_D(J_t, A_t, D_{t-1}, \epsilon_D). \tag{14}$$

Here $\{\epsilon_j | j \in \{N, P, J, A, E, D\}\}$ are learned from the data [1] . The data processing is following Aglietti et al. (2021). As shown in Figure, the three action nodes are $N_{in,t}$, $J_{t-1}$, and $A_{t-1}$. The intervention domains are $N_{in,t} \in [60.0, 100.0]$, $J_{t-1} \in [0.0, 36.0]$, and $A_{t-1} \in [0.0, 180.0]$. Here the domains are from the value range of the data. According to the result Figure 7, EXCBO outperforms the baselines on this real-world dataset.

### D.6 POOLED TESTING FOR COVID-19

We further compare EXCBO and existing methods using the COVID-19 pooled testing problem (Astudillo & Frazier, 2021a). The graphical structure is given by Figure 13-(c). In Figure 13-(c), $I_t$ is

---

[1] https://figshare.com/articles/dataset/Time_series_of_long-term_experimental_predator-prey_cycles/10045976/1

the fraction of the population that is infectious at time $t$; $R_t$ is the fraction of the population that is recovered and cannot be infected again, and time point $t \in \{1, 2, 3\}$. The additional fraction $S_t = 1 - I_t - R_t$ of the population is susceptible and can be infected. During each period $t$, the entire population is tested using a pool size of $x_t$. The loss $L_t$, incorporates the costs resulting from infections, testing resources used, and individuals isolated at period $t$. The objective is to choose pool size $x_t$ to minimize the total loss $\sum_t L_t$. Therefore, $x_t$s are the action variables/nodes that the algorithms try to optimize to achieve lower costs.

We employ the ANM (Hoyer et al., 2008) setup: $X_i = f(\mathbf{Z}_i) + U_i$, where $U_i \sim p(U) = 0.5\mathcal{N}(\mu_1, c_1\sigma^2) + 0.5\mathcal{N}(\mu_2, c_2\sigma^2)$, $c_1, c_2 > 0$. Data are generated using the dynamic SIR model from Astudillo & Frazier (2021a) with $\beta = 3.23$. For varying exogenous distributions $p(U)$, we use $\mu_1, \mu_2 \in [-0.5, 0.5]$ and $c_1, c_2 \in \{0.05, 0.5, 1.0\}$.

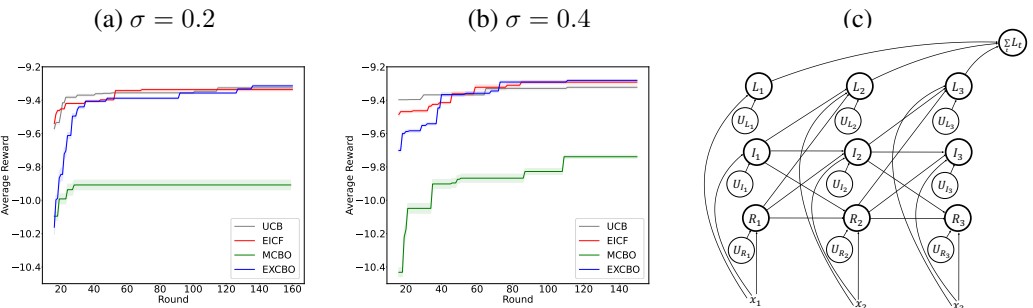

Figure 13: (a-b): Results of COVID-19 pooled testing optimization; (c): Graph structure for COVID-19 pooled testing problem.

Figure 13-(a-b) presents the optimization results obtained from different methods, where the reward is defined as $y = -\sum_t L_t$. As shown in Figure 13, UCB, EICF, and EXCBO exhibit similar performance across both $\sigma$ values. However, after 140 rounds, EXCBO achieves the best overall performance. The relatively poor performance of MCBO can be attributed partly to the bias introduced by the use of single-mode Gaussian distribution, and partly to the overfitting issues of the neural networks.

### D.7    RUNNING TIME

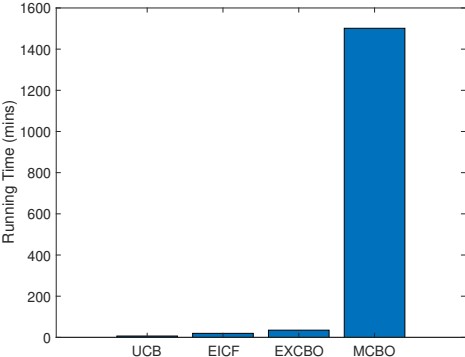

Figure 14: Running time of the algorithms on Dropwave data with $\sigma = 0.1$ and $\lambda = 1.0$ for four random seeds.

Table 1: Running time of the algorithms on Dropwave data with $\sigma = 0.1$ and $\lambda = 1.0$ for four random seeds.

| Methods | UCB | EICF | EXCBO | MCBO |
|---|---|---|---|---|
| Running Time (mins) | 6.5 | 19.6 | 35.1 | 1501.3 |

Figure 14 and Table 3 report the actual running time of the four algorithms on the Dropwave dataset with $\sigma = 0.1$ and $\lambda = 1.0$. Relative running times across datasets are consistent with the ratios shown in the figure. Empirically, EXCBO requires a similar amount of CPU time per iteration as UCB and EICF. In contrast, MCBO consumes significantly more computational resources — around 43 times as much — due to its reliance on neural networks. This highlights EXCBO's scalability advantage over existing state-of-the-art methods.

## D.8 EXCBO AND MCBO ON SINGLE-MODE EXOGENOUS DISTRIBUTION

We follow exactly the same setting in MCBO paper (Sussex et al., 2023) to compare EXCBO and MCBO using Dropwave data, i.e., $a_0, a_1 \in [0, 1]$, $X = \sqrt{(10.24a_0 - 5.12)^2 + (10.24a_1 - 5.12)^2}$, and $Y = (1.0 + \cos(12.0X))/(2.0 + 0.5X^2) + 0.1U, U \sim \mathcal{N}(0, 1)$, and the data generation code is from the MCBO package. The exogenous environment noise is unit-Gaussian scaled by 0.1. We report the best expected reward for both EXCBO and MCBO in Table 2. We can see EXCBO achieves improved performance in most steps, but MCBO gives a better result in the final round step $t = 100$.

Table 2: Results of Dropwave with unit-Gaussian noise.

| Round | 20 | 40 | 60 | 80 | 100 |
|---|---|---|---|---|---|
| MCBO | $0.78 \pm 0.05$ | $0.83 \pm 0.04$ | $0.87 \pm 0.03$ | $0.88 \pm 0.03$ | $0.91 \pm 0.02$ |
| EXCBO | $0.76 \pm 0.04$ | $0.84 \pm 0.04$ | $0.89 \pm 0.03$ | $0.89 \pm 0.02$ | $0.89 \pm 0.02$ |

Similarly, we follow the exact setting of Alphine2 in MCBO paper, i.e., $X_0 = -\sqrt{10.0a_0}\sin(10.0a_0) + U_0$, $X_i = \sqrt{10.0a_i}\sin(10.0a_i)X_{i-1} + U_i$ for $1 \leq i \leq 5$; and here $\mathbf{a}_i \in [0, 1], U_i \sim \mathcal{N}(0, 1), 0 \leq i \leq 5$. The exogenous environment noise is unit-Gaussian as reported in the MCBO paper. We report the best expected reward for both EXCBO and MCBO in Table 3.

Table 3: Results of Alphine2 with unit-Gaussian noise.

| Round | 20 | 40 | 60 | 80 | 100 |
|---|---|---|---|---|---|
| MCBO | $38.46 \pm 14.13$ | $76.47 \pm 16.56$ | $189.40 \pm 15.43$ | $327.07 \pm 12.38$ | $363.86 \pm 3.26$ |
| EXCBO | $28.98 \pm 13.32$ | $106.42 \pm 33.44$ | $166.48 \pm 42.43$ | $196.22 \pm 32.33$ | $241.57 \pm 14.00$ |

From these results, we conclude that for single-mode Gaussian exogenous distributions, MCBO performs better than EXCBO when the exogenous noise is strong (i.e., large $\sigma$, or large scale coefficient). In contrast, EXCBO achieves comparable or superior performance when the exogenous signal is weak or when $\sigma$ is small.

For multimodal exogenous distributions, as reported in Sections 7.4 and D.6, MCBO tends to be more vulnerable to complex exogenous distributions, particularly when they involve multimodal exogenous distributions with small variances. By comparison, the proposed exogenous learning framework effectively mitigates these challenges.

## D.9 ANALYSIS ON EXPERIMENTAL RESULTS

The experimental results across different datasets demonstrate that learning the exogenous distributions enhances EXCBO's ability to achieve optimal reward values. In particular, incorporating the distribution of exogenous variables yields a more accurate surrogate model when given an SCM and observational data.

Our method shows clear advantages over existing approaches when the exogenous noise is relatively weak. In such cases, the Gaussian Processes employed by UCB, EICF, and MCBO fail to capture the multimodality of the exogenous distribution, leading to a biased surrogate model with respect to the optimal intervention values. In contrast, EXCBO leverages a Gaussian mixture model, which effectively captures the multimodal exogenous distribution recovered by the proposed EDS under the DGM conditions. When the multimodal distribution of $U_i$ in $X_i = f(\mathbf{Z}_i, U_i)$ has small variances, the uncertainty is highly concentrated, making it harder to distinguish different modes in the plausible function map and resulting in larger bias in the objective approximation. By contrast, larger variances in the exogenous distribution allow the GPs in UCB, EICF, and MCBO to better discriminate between modes, thereby providing more accurate estimates of the expected objective function, i.e., $X_i = \mathbb{E}_{p(U_i)} f(\mathbf{Z}_i, U_i)$.

Gaps among different methods have been reported in previous studies, e.g., in MCBO (Sussex et al., 2023), Figures 2-f, 2-c, and 2-d. We speculate that this discrepancy arises because GPs with plain kernels are not universal approximators. Consequently, their limited expressiveness leads to irreducible bias, even with infinite data samples. This underscores the importance of incorporating structural knowledge to improve performance, as evidenced in MCBO, EICF, and EXCBO.

Finally, the regret bound in Theorem 6.1 depends on Assumptions 2–4 and holds with probability $1 - \alpha$, where $\alpha$ is specified in Assumption 4. This implies that different GP-based CBO methods are not guaranteed to converge to the same optimal reward value.

# E   PROOF OF THEOREM 4.1

Before we prove Theorem 4.1, we present a similar result for ANMs (Hoyer et al., 2008).

**Theorem E.1.** *Let $(X, \mathbf{Z}, U, f)$ be a node SCM. Let $\rho() : \mathcal{X} \times \mathcal{Z} \to \mathbb{R}^1$ be a predefined function regarding $X$ and $\mathbf{Z}$, and $\phi()$ be a regression model with $\phi() : \mathcal{Z} \to \rho(\mathcal{X}, \mathcal{Z})$. We define an encoder function $h() : \mathcal{Z} \times \mathcal{X} \to \widehat{\mathcal{U}}$ with $\widehat{U} := h(\mathbf{Z}, X) := \rho(X, \mathbf{Z}) - \phi(\mathbf{Z})$. The decoder is $g() : \mathcal{Z} \times \widehat{\mathcal{U}} \to \mathcal{X}$, i.e., $X = g(\mathbf{Z}, \widehat{U})$. Let $\rho()$ maps the values of $X$ and $\mathbf{Z}$ to an additive function of $\mathbf{Z}$ and $U$, i.e., $\rho(X, \mathbf{Z}) = \rho_1(\mathbf{Z}) + \rho_2(U)$. Then $\widehat{U} = h(\mathbf{Z}, X) = \rho_2(U) - \mathbb{E}[\rho_2(U)]$, and $\widehat{U} \perp\!\!\!\perp \mathbf{Z}$.*

*Proof.* As $\phi(\mathbf{z})$ is an optimal approximation of $\rho(X, \mathbf{z})$, with $\mathbf{Z} \perp\!\!\!\perp U$, we have

$$\phi(\mathbf{z}) = \mathbb{E}[\rho(X, \mathbf{z})] = \mathbb{E}[\rho_1(\mathbf{z}) + \rho_2(U)] = \int \big(\rho_1(\mathbf{z}) + \rho_2(u)\big) p(u) du$$

$$= \rho_1(\mathbf{z}) + \mathbb{E}[\rho_2(U)].$$

Thus, the decoder becomes

$$h(\mathbf{z}, x) = \rho(x, \mathbf{z}) - \phi(\mathbf{Z} = \mathbf{z})$$
$$= \rho_1(\mathbf{z}) + \rho_2(u) - \rho_1(\mathbf{z}) - \mathbb{E}[\rho_2(U)]$$
$$= \rho_2(u) - \mathbb{E}[\rho_2(U)].$$

Therefore, $\widehat{U} = h(\mathbf{Z}, X) = \rho_2(U) - \mathbb{E}[\rho_2(U)]$ is a function of $U$, and $h(\mathbf{Z}, X) \perp\!\!\!\perp \mathbf{Z}$, i.e., $\widehat{U} \perp\!\!\!\perp \mathbf{Z}$. □

*Example* 1. For an ANM (Hoyer et al., 2008) model $X = f(\mathbf{Z}) + U$, we have $\rho(X, \mathbf{Z}) = X$, $\rho_1(\mathbf{Z}) = f(\mathbf{Z})$, and $\rho_2(U) = U$, then $\widehat{U} = h(\mathbf{Z}, X) = U - \bar{U}$ .

*Example* 2. For a model $X = 2Ze^{-U} - e^{-Z}$, we have $\rho(X, Z) = \log(X + e^{-Z})$, $\rho_1(Z) = \log(2Z)$, and $\rho_2(U) = -U$, then $\widehat{U} = h(Z, X) = -U + \bar{U}$ .

Example 1 shows that the exogenous variable in any ANM model is identifiable. In practice, variable $X$'s generation mechanism $f()$ is generally unknown, and it is hard to propose a general form function $\rho()$ that can perform on any $f()$s and transform them to ANMs.

**Theorem 4.1** *Let $(\mathbf{Z}, U, X, f)$ be a node SCM, and $(\widehat{U}, \phi, h, g)$ an EDS surrogate of $U$. Suppose $f$ has the DGM structure, i.e. $X = f(\mathbf{Z}, U) = f_a(\mathbf{Z}) + f_b(\mathbf{Z}) f_c(U)$ with $f_b(\mathbf{z}) \neq 0$ for all $\mathbf{z} \in \mathcal{Z}$. In*

*addition, each $\mathbf{z} \in \mathcal{Z}$ has $N$ base samples in the close neighborhood of $\mathbf{z}$, i.e., $\left\{\mathbf{z}, x_i(\mathbf{z}, u_i)\right\}_{i=1}^{N}$ with $\{u_i\}_{i=1}^{N} \ i.i.d \sim p(U)$. Then with $N \to \infty$, the surrogate $\widehat{U} \to \frac{s}{\sigma_{f_c}}\left(f_c(U) - \mathbb{E}[f_c(U)]\right)$, $\mathbb{E}[\widehat{U}] \to 0$,* $\mathrm{Var}[\widehat{U}] \to 1$, *and $\widehat{U} \perp\!\!\!\perp \mathbf{Z}$, where $\sigma_{f_c} = \sqrt{\mathbb{E}\left[\left(f_c(U) - \mathbb{E}[f_c(U)]\right)^2\right]}$, $s \in \{-1, 1\}$.*

*Proof.* $\forall \mathbf{z} \in \mathcal{Z}$, with $\{u_i\}_{i=1}^{N} \ i.i.d \sim p(U)$, $N \to \infty$, and $\mathbf{Z} \perp\!\!\!\perp U$, the mean function is

$$\mu_\phi(\mathbf{z}) = \lim_{N \to \infty} \frac{1}{N} \sum_{i}^{N} x_i(\mathbf{z}, u_i) = \int \left(f_a(\mathbf{z}) + f_b(\mathbf{z})f_c(u)\right)p(u)du$$

$$= f_a(\mathbf{z}) + \int f_b(\mathbf{z})f_c(u)p(u)du$$

$$= f_a(\mathbf{z}) + f_b(\mathbf{z})\mathbb{E}\left[f_c(U)\right].$$

With mean $\mu_\phi(\mathbf{z})$, and an observation $x_i(\mathbf{z}, u_i)$, and $u_i \sim p(U)$,

$$\begin{aligned}
& x_i(\mathbf{z}, u_i) - \mu_\phi(\mathbf{z}) \\
=& f_a(\mathbf{z}) + f_b(\mathbf{z})f_c(u_i) - f_a(\mathbf{z}) - f_b(\mathbf{z})\mathbb{E}\left[f_c(U)\right] \\
=& f_b(\mathbf{z})\left(f_c(u_i) - \mathbb{E}[f_c(U)]\right).
\end{aligned} \tag{15}$$

With equation 15, $\forall \mathbf{z} \in \mathcal{Z}$, the variance of the regression model $\phi()$ is

$$\begin{aligned}
\sigma_\phi^2(\mathbf{z}) &= \lim_{N \to \infty} \frac{1}{N-1} \sum_{i=1}^{N} \left(x_i(\mathbf{z}, u_i) - \mu_\phi(\mathbf{z})\right)^2 \\
&= \lim_{N \to \infty} \frac{1}{N-1} \sum_{i=1}^{N} \left(f_b(\mathbf{z})\left(f_c(u_i) - \mathbb{E}[f_c(U)]\right)\right)^2 \\
&= f_b^2(\mathbf{z}) \lim_{N \to \infty} \frac{1}{N-1} \sum_{i=1}^{N} \left(f_c(u_i) - \mathbb{E}[f_c(U)]\right)^2 \\
&= f_b^2(\mathbf{z}) \int \left(f_c(u) - \mathbb{E}[f_c(U)]\right)^2 p(u)du \\
&= f_b^2(\mathbf{z})\mathbb{E}\left[\left(f_c(U) - \mathbb{E}[f_c(U)]\right)^2\right] \\
&= f_b^2(\mathbf{z})\sigma_{f_c}^2.
\end{aligned} \tag{16}$$

$\sigma_\phi^2(\mathbf{z})$ is the variance function with respect to variable $\mathbf{Z}$, i.e., $\sigma_\phi(\mathbf{z}) = \sigma_{f_c}|f_b(\mathbf{z})|$. Then, by equation 15 and equation 16, with $N \to \infty$,

$$\begin{aligned}
\frac{x - \mu_\phi(\mathbf{z})}{\sigma_\phi(\mathbf{z})} &= \frac{f_b(\mathbf{z})\left(f_c(u) - \mathbb{E}[f_c(U)]\right)}{\sigma_{f_c}|f_b(\mathbf{z})|} \\
&= \frac{s}{\sigma_{f_c}}\left(f_c(u) - \mathbb{E}[f_c(U)]\right).
\end{aligned} \tag{17}$$

Here $s = \mathrm{sign}[f_b(\mathbf{z})] \in \{1, -1\}$. As $f_b()$ is a continuous function, and $f_b(\mathbf{z}) \neq 0, \forall \mathbf{z} \in \mathcal{Z}$, $s = \mathrm{sign}[f_b(\mathbf{z})]$ is a constant value $\forall \mathbf{z} \in \mathcal{Z}$, either 1 or -1, and $s \perp\!\!\!\perp \mathbf{Z}$.

So with $N \to \infty$,

$$\widehat{U} = \frac{X - \mu_\phi(\mathbf{Z})}{\sigma_\phi(\mathbf{Z})} \to \frac{s}{\sigma_{f_c}}\left(f_c(U) - \mathbb{E}[f_c(U)]\right).$$

It shows that with $N \to \infty$, $\mathbb{E}[\widehat{U}] \to 0$, $\mathrm{Var}[\widehat{U}] \to 1$, and $\widehat{U} \perp\!\!\!\perp \mathbf{Z}$. $\qquad\square$

# F  EXOGENOUS DISTRIBUTION LEARNING

## F.1  CAUSAL INFERENCE WITH EXOGENOUS DISTRIBUTION

Under the monotonicity assumption on $f()$, the EDS framework can be extended to BGMs, building upon the analysis in (Lu et al., 2020; Nasr-Esfahany et al., 2023; Nasr-Esfahany & Kiciman, 2023; Chao et al., 2023). *Counterfactual* queries utilize functional models of generative processes to reason about alternative outcomes for individual data points, effectively answering questions like: "What if I had done A instead of B?" Such queries are formally described as a three-step process: abduction, action, and prediction (Pearl, 2009). A model that can be learned from data and execute these three steps is said to be *counterfactually identifiable*.

It is straightforward to show that a node SCM with a decomposable $f()$ is counterfactually identifiable. Thus, Theorem 4.1 introduces a novel class of node SCMs that achieve counterfactual identifiability beyond BGMs (Nasr-Esfahany et al., 2023).

*Remark* 1. We use the distribution of $\widehat{U} = \mathbf{s}(U) = h(\mathbf{Z}, X)$, i.e., $p(\widehat{U})$, to represent $p(U)$ within the surrogate model. With the decomposability assumption on $f()$, a node SCM is counterfactually identifiable.

Here, the parent set $\mathbf{Z}$ may include action variables, and the learned $\widehat{U}$ remains independent of the actions or interventions. Therefore, we can leverage the action variables to optimize the target variable through causal intervention operations.

This work lies within the line of research on counterfactual identification, such as ANM (Hoyer et al., 2008), BGM (Nasr-Esfahany et al., 2023), and LSNM (Immer et al., 2023). The proposed DGM is a new family of models that are counterfactually identifiable and can be easily implemented using GPs. Gaussian mixture models are employed to learn the recovered exogenous variable distribution, enabling a more accurate surrogate of the true data-generating mechanism, as demonstrated in the paper and our responses. The applicability of the proposed framework extends beyond CBO to broader causal inference tasks, including interventions and counterfactual inference.

## F.2  ANALYSIS ON BGMs

We first present a lemma on the BGM equivalence class of a node SCM with a monotonic mechanism.

**Lemma F.1.** *Let $(\mathbf{Z}, U, X, f)$ be a node SCM. $\forall \mathbf{z} \in \mathcal{Z}$, $f(\mathbf{z}, \cdot)$ is differentiable and strictly monotonic regarding $u \in \mathcal{U}$. We define a differentiable and invertible encoder function $h() : \mathcal{Z} \times \mathcal{X} \to \widehat{\mathcal{U}}$, i.e., $\widehat{U} := h(\mathbf{Z}, X)$, and $\widehat{U} \perp\!\!\!\perp \mathbf{Z}$. The decoder is $g() : \mathcal{Z} \times \widehat{\mathcal{U}} \to \mathcal{X}$, i.e., $X = g(\mathbf{Z}, \widehat{U})$. Then $\widehat{U} = h(\mathbf{Z}, X)$ is a function of $U$, i.e., $\widehat{U} = \mathbf{s}(U)$, and $\mathbf{s}()$ is an invertible function.*

*Proof.* According to the definition of node SCM, we have $\mathbf{Z} \perp\!\!\!\perp U$. According to the assumption, $\forall \mathbf{z} \in \mathcal{Z}$, $f(\mathbf{z}, u)$ is differentiable and strictly monotonic regarding $u$. Hence $X = f(\mathbf{Z}, U)$ is a BGM, and we use $\mathbb{F}$ to represent BGM class that satisfies the independence ($\mathbf{Z} \perp\!\!\!\perp U$) and the function monotone conditions. We can see that $h^{-1} \in \mathbb{F}$, $h^{-1}(\mathbf{z}, \cdot) = g(\mathbf{z}, \cdot)$, and $h^{-1}(\mathbf{z}, \cdot)$ and $f(\mathbf{z}, \cdot)$ are counterfactually equivalent BGMs that generate the same distribution $p(\mathbf{Z}, X)$. Based Lemma B.2, Proposition 6.2, and Definition 6.1 in (Nasr-Esfahany et al., 2023), there exists an invertible function $\mathbf{s}()$ that satisfies $\forall \mathbf{z} \in \mathcal{Z}, x \in \mathcal{X}, h(\mathbf{z}, x) = \mathbf{s}(f^{-1}(\mathbf{z}, x))$, i.e., $\widehat{u} = h(\mathbf{z}, x) = \mathbf{s}(f^{-1}(\mathbf{z}, x)) = \mathbf{s}(u)$, which is $\widehat{U} = \mathbf{s}(U)$. $\qquad\square$

We can easily prove that an EDS model of a monotonic node SCM belongs to its BGM equivalence class under the independence assumption $\widehat{U} \perp\!\!\!\perp \mathbf{Z}$.

**Theorem F.2.** *Let $(\mathbf{Z}, U, X, f)$ be a node SCM. $\forall \mathbf{z} \in \mathcal{Z}$, $f(\mathbf{z}, \cdot)$ is differentiable and strictly monotonic regarding $u \in \mathcal{U}$. Let $(\widehat{U}, \phi, h, g)$ be an EDS surrogate of $U$. We further assume that $\widehat{U} \perp\!\!\!\perp \mathbf{Z}$. Then $\widehat{U} = h(\mathbf{Z}, X)$ is a function of $U$, i.e., $\widehat{U} = \mathbf{s}(U)$, and $\mathbf{s}()$ is an invertible function.*

*Proof.* It is to prove that the encoder of an EDS, i.e., $\widehat{U} = h(\mathbf{Z}, X) = \frac{X - \mu_\phi(\mathbf{Z})}{\sigma_\phi(\mathbf{Z})}$, is invertible regarding $\widehat{U}$ and $X$ given a value of $\mathbf{Z}$. With the assumption $\widehat{U} \perp\!\!\!\perp \mathbf{Z}$, by using the results of

Lemma F.1, we have $\widehat{U} = h(\mathbf{Z}, X)$ is a function of $U$, i.e., $\widehat{U} = \mathbf{s}(U)$, and $\mathbf{s}()$ is an invertible function. □

Based on the proof of Theorem F.2, a node SCM with a monotonic mechanism is counterfactually identifiable by using an EDS model with the $\widehat{U} \perp\!\!\!\perp \mathbf{Z}$ constraint.

# G   REGRET ANALYSIS

## G.1   REMARKS ON REGRET BOUND

The analysis in this paper focuses on the DGM mechanisms. To extend the analysis to BGMs, we need to consider the computation cost involving the independence penalization on variables $\widehat{U}$ and $\mathbf{Z}$. For mechanisms beyond DGMs and BGMs, we conjecture that the surrogate approximation accuracy may decrease, but the convergence rate may not decrease a lot. The cumulative regret provides insight into the convergence behavior of the algorithm.

Our analysis follows the study in (Sussex et al., 2023). In the DAG $\mathcal{G}$ over $\{X_i\}_0^d$, let $N$ be the maximum distance from a root to $X_d$, i.e., $N = \max_i \text{dist}(X_i, X_d)$. Here $\text{dist}(\cdot, \cdot)$ is a measure of the edges in the longest path from $X_i$ to the reward node $X_d$. Let $M$ denote the maximum number of parents of any variables in $\mathcal{G}$, $M = \max_i |\mathbf{pa}(i)|$. Let $L_t$ be a function of $L_g, L_{\sigma_g}$. According to Theorem 4.1, with the EDS structure given in Figure 2 in the main text, the exogenous variable and its distribution can be recovered. For each observation of the dynamic surrogate model, we assume the sampling of $p(\widehat{U})$, $\widehat{\tilde{u}} = \mathbf{s}(\tilde{u}) = \mathbf{s}(u)$. This maximum information gain is commonly used in many Bayesian Optimizations (Srinivas et al., 2010). Many common kernels, such as linear and squared exponential kernels, lead to sublinear information gain in $T$, and it results in an overall sublinear regret for EXCBO (Sussex et al., 2023).

## G.2   PROOF OF THEOREM 6.1

We give the assumptions used in the regret analysis. Assumption 2 gives the Lipschitz conditions of $g_i, \sigma_{g,i}$, and $\mu_{g,i}$. It holds if the RKHS of each $g_i$ has a Lipschitz continuous kernel (Curi et al., 2020; Sussex et al., 2023). Assumption 4 holds when we assume that the $i$th GP prior uses the same kernel as the RKHS of $g_i$ and that $\beta_{i,t}$ is sufficiently large to ensure the confidence bounds in

$$\left| g_i(\mathbf{z}_i, \mathbf{a}_i, \widehat{u}_i) - \mu_{g,i,t-1}(\mathbf{z}_i, \mathbf{a}_i, \widehat{u}_i) \right| \leq \beta_{i,t} \sigma_{g,i,t-1}(\mathbf{z}_i, \mathbf{a}_i, \widehat{u}_i), \quad \forall \mathbf{z}_i \in \mathcal{Z}_i, \mathbf{a}_i \in \mathbf{A}_i, \widehat{u}_i \in \widehat{\mathcal{U}}_i.$$

**Assumption 2.** $\forall g_i \in \mathbf{G}$, $g_i$ is $L_g$-Lipschitz continuous; moreover, $\forall i, t$, $\mu_{g,i,t}$ and $\sigma_{g,i,t}$ are $L_{\mu_g}$ and $L_{\sigma_g}$ Lipschitz continuous.

**Assumption 3.** $\forall f_i \in \mathbf{F}$, $f_i$ is differentiable and has a decomposable structure with $X = f_i(\mathbf{Z}_i, U_i) = f_{i(a)}(\mathbf{Z}_i) + f_{i(b)}(\mathbf{Z}_i)f_{i(c)}(U_i)$, and $f_{i(b)}(\mathbf{z}_i) \neq 0, \forall \mathbf{z}_i \in \mathcal{Z}_i$.

**Assumption 4.** $\forall i, t$, there exists sequence $\beta_{i,t} \in \mathbb{R}_{>0}$, with probability at least $(1 - \alpha)$, for all $\mathbf{z}_i, \mathbf{a}_i, \widehat{u}_i \in \mathcal{Z}_i \times \mathbf{A}_i \times \widehat{\mathcal{U}}_i$ we have $\left| g_i(\mathbf{z}_i, \mathbf{a}_i, \widehat{u}_i) - \mu_{g,i,t-1}(\mathbf{z}_i, \mathbf{a}_i, \widehat{u}_i) \right| \leq \beta_{i,t} \sigma_{g,i,t-1}(\mathbf{z}_i, \mathbf{a}_i, \widehat{u}_i)$, and $|h(\mathbf{z}_i, \mathbf{a}_i, x_i) - \mu_{h,i,t-1}(\mathbf{z}_i, \mathbf{a}_i, x_i)| \leq \beta_{i,t} \sigma_{h,i,t-1}(\mathbf{z}_i, \mathbf{a}_i, x_i)$.

Following Chowdhury & Gopalan (2019), at time $t$, let $\tilde{\mathbf{G}}$ be the statistically plausible function set of $\mathbf{G}$, i.e., $\tilde{\mathbf{G}} = \{\tilde{g}_i\}_{i=0}^d$. The following lemma bounds the value of $\widehat{\tilde{u}}$ with the variance of the encoder.

**Lemma G.1.**
$$\|\widehat{u}_{i,t} - \widehat{\tilde{u}}_{i,t}\| \leq 2\beta_t \|\sigma_{\widehat{u}_{i,t-1}}\| = 2\beta_t \|\sigma_{h,i,t-1}\|.$$

*Proof.* With Assumption 4 and $\widehat{u}_{i,t} = h_{i,i-1}(\mathbf{z}_i, \mathbf{a}_i, x_i)$, let $\widehat{\tilde{u}}_{i,t} = \mu_{\widehat{u}_{i,t-1}}\mathbf{z}_i, \mathbf{a}_i, x_i + \beta_t \sigma_{\widehat{u}_{i,t-1}}(\mathbf{z}_i, \mathbf{a}_i, x_i) \circ \boldsymbol{\omega}_{\widehat{u}_{i,t-1}}(\mathbf{z}_i, \mathbf{a}_i, x_i)$, and here $|\boldsymbol{\omega}_{\widehat{u}_{i,t-1}}(\mathbf{z}_i, \mathbf{a}_i, x_i)| \leq 1$. Then

$$\begin{aligned}
\|\widehat{u}_{i,t} - \widehat{\tilde{u}}_{i,t}\| =& \|\widehat{\tilde{u}}_{i,t} - \mu_{\widehat{u}_{i,t-1}}(\mathbf{z}_i, \mathbf{a}_i, x_i) - \beta_t \sigma_{\widehat{u}_{i,t-1}}(\mathbf{z}_i, \mathbf{a}_i, x_i) \circ \boldsymbol{\omega}_{\widehat{u}_{i,t-1}}(\mathbf{z}_i, \mathbf{a}_i, x_i)\| \\
\leq& \|\widehat{\tilde{u}}_{i,t} - \mu_{\widehat{u}_{i,t-1}}(\mathbf{z}_i, \mathbf{a}_i, x_i)\| + \beta_t \|\sigma_{\widehat{u}_{i,t-1}}(\mathbf{z}_i, \mathbf{a}_i, x_i) \circ \boldsymbol{\omega}_{\widehat{u}_{i,t-1}}(\mathbf{z}_i, \mathbf{a}_i, x_i)\| \\
\leq& 2\beta_t \|\sigma_{\widehat{u}_{i,t-1}}(\mathbf{z}_i, \mathbf{a}_i, x_i)\| = 2\beta_t \|\sigma_{h,i,t-1}\|.
\end{aligned}$$

□

With the decomposable Assumption 3 on $f_i$, $\sigma^2_{h,i,t-1} \propto f^2_{i(b)}(\mathbf{z}_i, \mathbf{a}_i)\big(f_{i(c)}(U) - \mathbb{E}[f_{i(c)}(U)]\big)^2$ according to the proof of Theorem 4.1. $f_{i(b)}()$ is learned with the variance of regression model $\phi()$, i.e. $\sigma_{\phi,i,t}()$.

**Lemma G.2.**

$$\|x_{d,t} - \tilde{x}_{d,t}\| \le 2\beta_t M^{N_i}(2\beta_t L_{\sigma_g} + L_g)^{N_i} \sum_{j=0}^{i} \big(\sigma_{g,j,t-1}(\mathbf{z}_{j,t}) + \sigma_{\widehat{u}_{j,t-1}}\big).$$

*Proof.* We use $g_i(\mathbf{z}_{i,t}, \widehat{u}_{i,t})$ to represent $g_i(\mathbf{z}_{i,t}, \mathbf{a}_{i,t}, \widehat{u}_{i,t})$ because we assume the actions to be the same for the process generating $x_{i,t}$ and $\tilde{x}_{i,t}$. Similarly, $\mu_{g,i,t-1}(\tilde{\mathbf{z}}_{i,t}, \tilde{\widehat{u}}_{i,t}) = \mu_{g,i,t-1}(\tilde{\mathbf{z}}_{i,t}, \tilde{\mathbf{a}}_{i,t}, \tilde{\widehat{u}}_{i,t})$, $\sigma_{g,i,t-1}(\tilde{\mathbf{z}}_{i,t}, \tilde{\widehat{u}}_{i,t}) = \sigma_{g,i,t-1}(\tilde{\mathbf{z}}_{i,t}, \tilde{\mathbf{a}}_{i,t}, \tilde{\widehat{u}}_{i,t})$.

We use the reparameterization trick, and write $\tilde{x}_{i,t}$ as

$$\tilde{x}_{i,t} = \tilde{g}_i(\tilde{\mathbf{z}}_{i,t}, \tilde{\widehat{u}}_{i,t}) = \mu_{g,i,t-1}(\tilde{\mathbf{z}}_{i,t}, \tilde{\widehat{u}}_{i,t}) + \beta_t \sigma_{g,i,t-1}(\tilde{\mathbf{z}}_i, \tilde{\widehat{u}}_{i,t}) \circ \boldsymbol{\omega}_{g,i,t-1}(\tilde{\mathbf{z}}_i, \tilde{\widehat{u}}_{i,t}).$$

Here $|\boldsymbol{\omega}_{g,i,t-1}(\tilde{\mathbf{z}}_i, \tilde{\widehat{u}}_{i,t})| \le 1$. Hence, we have

$$\|x_{i,t} - \tilde{x}_{i,t}\| = \|g_i(\mathbf{z}_{i,t}, \widehat{u}_{i,t}) - \mu_{g,i,t-1}(\tilde{\mathbf{z}}_{i,t}, \tilde{\widehat{u}}_{i,t}) - \beta_t \sigma_{g,i,t-1}(\tilde{\mathbf{z}}_i, \tilde{\widehat{u}}_{i,t})\boldsymbol{\omega}_{g,i,t-1}(\tilde{\mathbf{z}}_i, \tilde{\widehat{u}}_{i,t})\|$$

$$= \|g_i(\tilde{\mathbf{z}}_{i,t}, \tilde{\widehat{u}}_{i,t}) - \mu_{g,i,t-1}(\tilde{\mathbf{z}}_{i,t}, \tilde{\widehat{u}}_{i,t}) - \beta_t \sigma_{g,i,t-1}(\tilde{\mathbf{z}}_i, \tilde{\widehat{u}}_{i,t})\boldsymbol{\omega}_{g,i,t-1}(\tilde{\mathbf{z}}_i, \tilde{\widehat{u}}_{i,t})$$

$$\qquad + g_i(\mathbf{z}_{i,t}, \widehat{u}_{i,t}) - g_i(\tilde{\mathbf{z}}_{i,t}, \tilde{\widehat{u}}_{i,t})\|$$

$$\le \|g_i(\tilde{\mathbf{z}}_{i,t}, \tilde{\widehat{u}}_{i,t}) - \mu_{g,i,t-1}(\tilde{\mathbf{z}}_{i,t}, \tilde{\widehat{u}}_{i,t})\| + \|\beta_t \sigma_{g,i,t-1}(\tilde{\mathbf{z}}_i, \tilde{\widehat{u}}_{i,t})\boldsymbol{\omega}_{g,i,t-1}(\tilde{\mathbf{z}}_i, \tilde{\widehat{u}}_{i,t})\|$$

$$\qquad + \|g_i(\mathbf{z}_{i,t}, \widehat{u}_{i,t}) - g_i(\tilde{\mathbf{z}}_{i,t}, \tilde{\widehat{u}}_{i,t})\|$$

$$\overset{\zeta_1}{\le} \beta_t \|\sigma_{g,i,t-1}(\tilde{\mathbf{z}}_i, \tilde{\widehat{u}}_{i,t})\| + \beta_t \|\sigma_{g,i,t-1}(\tilde{\mathbf{z}}_i, \tilde{\widehat{u}}_{i,t})\| + L_{g_i}\big\|[\mathbf{z}_{i,t}; \widehat{u}_{i,t}] - [\tilde{\mathbf{z}}_{i,t}; \tilde{\widehat{u}}_{i,t}]\big\|$$

$$= 2\beta_t \|\sigma_{g,i,t-1}(\mathbf{z}_i, \widehat{u}_{i,t}) + \sigma_{g,i,t-1}(\tilde{\mathbf{z}}_i, \tilde{\widehat{u}}_{i,t}) - \sigma_{g,i,t-1}(\mathbf{z}_i, \widehat{u}_{i,t})\| + L_{g_i}\big\|[\mathbf{z}_{i,t}; \widehat{u}_{i,t}] - [\tilde{\mathbf{z}}_{i,t}; \tilde{\widehat{u}}_{i,t}]\big\|$$

$$\overset{\zeta_2}{\le} 2\beta_t\Big(\|\sigma_{g,i,t-1}(\mathbf{z}_i, \widehat{u}_{i,t})\| + L_{\sigma_{g,i}}\big\|[\mathbf{z}_{i,t}; \widehat{u}_{i,t}] - [\tilde{\mathbf{z}}_{i,t}; \tilde{\widehat{u}}_{i,t}]\big\|\Big) + L_{g_i}\big\|[\mathbf{z}_{i,t}; \widehat{u}_{i,t}] - [\tilde{\mathbf{z}}_{i,t}; \tilde{\widehat{u}}_{i,t}]\big\|$$

$$= 2\beta_t \sigma_{g,i,t-1}(\mathbf{z}_i, \widehat{u}_{i,t}) + (2\beta_t L_{\sigma_{g,i}} + L_{g_i})\big\|[\mathbf{z}_{i,t}; \widehat{u}_{i,t}] - [\tilde{\mathbf{z}}_{i,t}; \tilde{\widehat{u}}_{i,t}]\big\|$$

$$\le 2\beta_t \sigma_{g,i,t-1}(\mathbf{z}_i, \widehat{u}_{i,t}) + (2\beta_t L_{\sigma_{g,i}} + L_{g_i})\|\mathbf{z}_{i,t} - \tilde{\mathbf{z}}_{i,t}\| + (2\beta_t L_{\sigma_{g,i}} + L_{g_i})\|\widehat{u}_{i,t} - \tilde{\widehat{u}}_{i,t}\|$$

$$\overset{\zeta_3}{\le} 2\beta_t \sigma_{g,i,t-1}(\mathbf{z}_i, \widehat{u}_{i,t}) + (2\beta_t L_{\sigma_{g,i}} + L_{g_i})\|\mathbf{z}_{i,t} - \tilde{\mathbf{z}}_{i,t}\| + 2\beta_t(2\beta_t L_{\sigma_{g,i}} + L_{g_i})\sigma_{\widehat{u}_{i,t-1}}$$

$$= 2\beta_t \sigma_{g,i,t-1}(\mathbf{z}_i, \widehat{u}_{i,t}) + 2\beta_t(2\beta_t L_{\sigma_{g,i}} + L_{g_i})\sigma_{\widehat{u}_{i,t-1}} + (2\beta_t L_{\sigma_{g,i}} + L_{g_i})\sum_{j\in\mathbf{pa}(i)}\|\mathbf{z}_{j,t} - \tilde{\mathbf{z}}_{j,t}\|$$

$$\le 2\beta_t \sigma_{g,i,t-1}(\mathbf{z}_i, \widehat{u}_{i,t}) + 2\beta_t(2\beta_t L_{\sigma_g} + L_g)\sigma_{\widehat{u}_{i,t-1}} + (2\beta_t L_{\sigma_g} + L_g)\sum_{j\in\mathbf{pa}(i)}\|x_{j,t} - \tilde{x}_{j,t}\|$$

$$\overset{\zeta_4}{\le} 2\beta_t \sigma_{g,i,t-1}(\mathbf{z}_i, \widehat{u}_{i,t}) + 2\beta_t(2\beta_t L_{\sigma_g} + L_g)\sigma_{\widehat{u}_{i,t-1}}$$

$$\qquad + (2\beta_t L_{\sigma_g} + L_g)\sum_{j\in\mathbf{pa}(i)} 2\beta_t M^{N_j}(2\beta_t L_{\sigma_g} + L_g)^{N_j}\sum_{h=0}^{j}\big(\sigma_{g,h,t-1}(\mathbf{z}_{h,t}) + \sigma_{\widehat{u}_{h,t-1}}\big)$$

$$\le 2\beta_t M^{N_i}(2\beta_t L_{\sigma_g} + L_g)^{N_i}\sum_{j=0}^{i}\big(\sigma_{g,j,t-1}(\mathbf{z}_{j,t}) + \sigma_{\widehat{u}_{j,t-1}}\big)$$

In steps $\zeta_1$ and $\zeta_2$, we rely on the calibrated uncertainty and Lipschitz dynamics; in step $\zeta_2$, we also apply the triangle inequality; step $\zeta_3$ is by Lemma G.1; $\zeta_4$ applies the inductive hypothesis. $\qquad\square$

**Theorem 6.1** *Consider the optimization problem in equation 3, with the SCM satisfying Assumptions 2- 4, where $\mathcal{G}$ is known but $\mathbf{F}$ is unknown. Then with probability at least $1 - \alpha$, the cumulative regret of Algorithm 1 is bounded by*

$$R_T \le \mathcal{O}(L_T M^N d\sqrt{T\gamma_T}).$$

*Proof.* The cumulative regret is

$$R_T = \sum_{t=1}^{T} \left[ \mathbb{E}[y|\mathbf{a}^*] - \mathbb{E}[y|\mathbf{a}_{:,t}] \right].$$

At step $t$, the instantaneous regret is $r_t$. By applying Lemma G.2, $r_t$ is bounded by

$$\begin{aligned}
r_t &= \mathbb{E}[y|\mathbf{F}, \mathbf{a}^*] - \mathbb{E}[y|\mathbf{F}, \mathbf{a}_{:,t}] \\
&\leq \mathbb{E}[y_t|\tilde{\mathbf{F}}, \mathbf{a}_{:,t}] - \mathbb{E}[y_t|\mathbf{F}, \mathbf{a}_{:,t}] \\
&= \mathbb{E}[\|x_{i,t} - \tilde{x}_{i,t}\| | \mathbf{a}_{:,t}] \\
&\leq 2\beta_t M^N (2\beta_t L_{\sigma_g} + L_g)^N \mathbb{E}\left[ \sum_{i=0}^{d} \|\sigma_{g,i,t-1}(\mathbf{z}_{i,t})\| + \|\sigma_{\widehat{u}_{i,t-1}}\| \right]
\end{aligned}$$

Here $L_t = 2\beta_t (2\beta_t L_{\sigma_g} + L_g)^N$. Thus,

$$\begin{aligned}
r_t^2 &\leq L_t^2 M^{2N} \left( \mathbb{E}\left[ \sum_{i=0}^{d} \|\sigma_{g,i,t-1}(\mathbf{z}_{i,t})\| + \|\sigma_{\widehat{u}_{i,t-1}}\| \right] \right)^2 \\
&\leq 2d L_t^2 M^{2N} \mathbb{E}\left[ \sum_{i=0}^{d} \|\sigma_{g,i,t-1}(\mathbf{z}_{i,t})\|_2^2 + \|\sigma_{\widehat{u}_{i,t-1}}\|_2^2 \right]
\end{aligned}$$

We define $R_T^2$ as

$$\begin{aligned}
R_T^2 &= \left( \sum_{t=1}^{T} r_t \right)^2 \leq T \sum_{t=1}^{T} r_t^2 \\
&\leq 2dT L_T^2 M^{2N} \sum_{t=1}^{T} \mathbb{E}\left[ \sum_{i=0}^{d} \|\sigma_{g,i,t-1}(\mathbf{z}_{i,t})\|_2^2 + \|\sigma_{\widehat{u}_{i,t-1}}\|_2^2 \right] \\
&= 2dT L_T^2 M^{2N} \Gamma_T.
\end{aligned}$$

Here,

$$\begin{aligned}
\Gamma_T &= \max_{(\mathbf{z},\mathbf{a},\widehat{u}) \in \mathcal{Z} \times \mathcal{A} \times \widehat{\mathcal{U}}} \sum_{t=1}^{T} \sum_{i=0}^{d} \left[ \|\sigma_{i,t-1}(\mathbf{z}_{i,t}, \mathbf{a}_{i,t})\|_2^2 + \|\sigma_{\widehat{u}_{i,t-1}}\|_2^2 \right] \\
&\leq \max_{\mathbf{A},\widehat{U}} \sum_{t=1}^{T} \sum_{i=0}^{d} \left[ \|\sigma_{i,t-1}(\mathbf{z}_{i,t}, \mathbf{a}_{i,t})\|_2^2 + \|\sigma_{\widehat{u}_{i,t-1}}\|_2^2 \right] \\
&\leq \sum_{i=0}^{d} \max_{\mathbf{A}_i,\widehat{U}_i} \sum_{t=1}^{T} \left[ \|\sigma_{i,t-1}(\mathbf{z}_{i,t}, \mathbf{a}_{i,t})\|_2^2 + \|\sigma_{\widehat{u}_{i,t-1}}\|_2^2 \right] \\
&\leq \sum_{i=0}^{d} \max_{\mathbf{A}_i,\widehat{U}_i} \sum_{t=1}^{T} \left[ \sum_{l=1}^{d_i} \|\sigma_{i,t-1}(\mathbf{z}_{i,t}, \mathbf{a}_{i,t}, l)\|_2^2 + \|\sigma_{\widehat{u}_{i,t-1}}\|_2^2 \right] \\
&\overset{\zeta_1}{\leq} \sum_{i=0}^{d} \frac{2}{\ln(1 + \rho_i^{-2})} \gamma_{i,T} \\
&= \mathcal{O}(d\gamma_T).
\end{aligned}$$

Here $\zeta_1$ is due to the upper bound of the information gain (Srinivas et al., 2010), and $\gamma_T$ will often scale sublinearly in $T$ (Sussex et al., 2023). Therefore,

$$R_T^2 \leq 2T L_T^2 M^{2N} d \mathcal{O}(d\gamma_T).$$

And,

$$R_T \leq \mathcal{O}(L_T M^N d \sqrt{T\gamma_T}).$$

This completes the proof of the theorem.

$\square$

