# OpenReview forum: "Exogenous Distribution Learning for Causal Bayesian Optimization"
_ICLR.cc/2026/Conference — Submitted to ICLR 2026_

### Official Review · Reviewer_V2tA · 2025-10-31

**Soundness:** 2
**Presentation:** 2
**Contribution:** 2
**Rating:** 2
**Confidence:** 3

**Summary:**

This paper proposes EXCBO, a Causal Bayesian Optimization method that explicitly learns the distribution of exogenous noise variables in structural causal models rather than marginalizing them out. The authors use an encoder-decoder framework to recover exogenous variables from observational data and model their distribution with Gaussian Mixture Models. They introduce the Decomposable Generation Mechanism (DGM) as a generalization of Additive Noise Models (ANM), prove exogenous recovery under DGM, and provide regret analysis. Experiments on synthetic and real-world datasets show EXCBO can outperform baselines when noise is multimodal with moderate variance.

**Strengths:**

1. This paper addresses a gap in existing CBO methods by explicitly modeling exogenous distributions rather than marginalizing them out or assuming simple Gaussian noise.

2. The encoder-decoder framework is intuitive and practical, using standardization to recover exogenous variables from observational data. The DGM formulation is more general than the ANM.

**Weaknesses:**

1. The authors use 2-component Gaussian Mixture Models to model $p(\hat{U})$ without any justification. Could the authors elaborate on a) why GMMs were chosen over other flexible density estimators? b) Why exactly 2 components? c) How sensitive are the results to this choice?

2. Regarding the theoretical analysis part:

a) The authors mention that there exists a constant $a$ in Theorem 4.1, but according to Equation (16), $a = \mathrm{sign}[f_b(z)/c]$ actually depends on $z$. Please clarify whether a is truly constant or the independence claim needs modification.

b) For BGM (Theorem F.2), $\hat{U} \perp\perp Z$ is explicitly assumed as a premise. For DGM (Theorem 4.1), it is claimed to be proven as a conclusion. However, the DGM proof relies on $a$ being constant (which is a concern in 2(a)).

c) In line 1037, the authors stated that $\sigma_{\phi}(z) = c|f_b(z)|$. Can the authors elaborate on whether this equation is correct?

3. Regarding the experiments:

a) MCBO is missed in the Dropwave experiments in Figure 4. I wonder if the authors could provide justification for it? From Figure 9, it seems that MCBO performs comparably with EXCBO on the Dropwave experiments.

b) Figure 4 shows EXCBO's advantage decreases as $\lambda$ and $\sigma$ increase. Can the authors elaborate more on it and provide more explanation?

4. The paper is notation-heavy with many symbols, which makes it a little bit hard to understand and follow. It would be better if the authors could add more illustrative examples to improve clarity

5. (Minor) The indentation and margin of the beginning of many paragraphs (e.g., the first paragraph in Section 3, Section 3.4) should be adjusted.

**Questions:**

Please see the questions in the Weaknesses part.

---

> ### Author Response · Authors · 2025-12-04
>
> We thank the reviewer for their valuable feedback and for taking the time to assess our work. Below, we address each concern point by point.
>
> **1. Two-component Gaussian Mixture Models**
>
> The synthetic datasets are generated using two-component noise for each node, which motivates our choice of using two components in the learning model. Since GMMs are universal approximators capable of modeling any distribution given a sufficient number of components, we additionally provide experiments in Figure 9 to demonstrate the robustness of our method to different component choices. The results in Figure 9 show that EXCBO performs similarly across different component numbers, as long as the number of components is greater than the true number of modes in the noise.
>
> **2. Theoretical analysis**
>
> a) Since $f_b()$ is a continuous function and $f_b(\mathbf{z}) \neq 0, \forall \mathbf{z} \in \mathcal{Z}$, the value $s = \textrm{sign}[f_b(\mathbf{z})]$ is a constant, either $1$ or $-1$, and thus $s$ is independent of $\mathbf{Z}$. We have revised the proof to make the argument clearer.
>
> b) The variable $\widehat{U}$ serves as a surrogate for $U$ and is learned from data. The condition $\widehat{U}$ independent of $\mathbf{Z}$ is required to ensure counterfactual identifiability of the model.
>
> c) We have revised the proof of Theorem 4.1 to improve clarity and readability.
>
> **3. Experiments**
>
> We have added MCBO results to the plots for Dropwave and Alpine2. MCBO and EXCBO achieve similar performance on Dropwave, which is expected due to the structural simplicity of the Dropwave function.
>
> **4. Notation**
>
> We have updated the notation table to include additional symbols and simplified several notations in the main text to improve readability.

---

### Official Review · Reviewer_G7dR · 2025-10-31

**Soundness:** 2
**Presentation:** 3
**Contribution:** 2
**Rating:** 4
**Confidence:** 4

**Summary:**

The paper introduces EXCBO, a causal Bayesian optimization framework that relaxes the restrictive additive Gaussian noise assumption by estimating the exogenous variable distribution from data.
Using an encoder decoder surrogate (EDS), EXCBO recovers latent residuals, models their distribution with a Gaussian Mixture Model, and integrates the estimated p(U) into the optimization process.
Empirical results on synthetic and real-world structural causal models show improvements over standard CBO baselines under multimodal or non-Gaussian noise.

**Strengths:**

Strengths
Clear motivation: Identifies a genuine limitation in prior CBO frameworks that assume additive Gaussian noise, a well-motivated problem in causal optimization.

Intuitive methodology: The encoder-decoder formulation for exogenous recovery is conceptually sound and connects structural causal modeling with modern regression techniques.

Reasonable empirical evidence: Experiments demonstrate that EXCBO improves performance in settings with multimodal or non-Gaussian noise distributions, aligning with its theoretical motivation.

Practical relevance: Learning a more realistic noise model can be useful for real-world decision-making tasks

**Weaknesses:**

Weaknesses
Conceptual and Theoretical
Incremental novelty: The paper combines known ideas from heteroscedastic Gaussian Processes, nonlinear ICA, and latent-variable Bayesian optimization rather than introducing fundamentally new theory or algorithms.

Limited theoretical depth: The recoverability theorem is a restatement of standard residual properties under independence. The regret bound simply inherits results from GP-UCB without considering estimation uncertainty from the exogenous step.

No analysis of identifiability or robustness: The paper does not explore what happens when the DGM assumption fails, when noise is correlated with parents, or when the graph is misspecified.
Algorithmic
Minor procedural change: The algorithm is essentially FNBO plus a residual normalization and GMM fitting step. The “learning” component is non-iterative and computed once before optimization.

Omission of MCBO: The most relevant baseline (MCBO) is missing from Figures 4 and 5. The justification (“computationally expensive”) is weak and unsupported by runtime data.

Unexplained high initial rewards: In the reward progression plots, EXCBO starts significantly higher than other methods. The paper does not explain whether EXCBO uses pretraining or a different initialization, raising concerns about comparability.

Experimental
Inconsistent reporting: The number of experimental runs or seeds is clearly stated (four) only for the Dropwave dataset. Other benchmarks have uncertainty bars but no run count.

Limited noise diversity: Only Gaussian and two-component Gaussian mixture noises are tested. There are no experiments with heavy-tailed, skewed, or heteroscedastic noise beyond the DGM structure.

No robustness or ablation studies: The effects of the encoder-decoder, the GMM modeling, or independence violations are not separately tested.

Partial tabular reporting: Tables 1 and 2 summarize results for small-scale tasks, but not for larger experiments or real-world cases.

Writing / Presentation
Clear overall, but contains typographical errors ( “STATMENT”, “LLMS”, extra parenthesis in “do(XI := f(ZI, A, UI) )”.

Related works is missing critical prior work related to Heteroskedastic Gaussian Processes, Latent Variable Bayesian Optimisation and Non Linear ICA, all closely related to this work

Minor formatting inconsistencies in math expressions and figures.

 Minor Comments
Ensure consistent reporting of the number of runs/seeds across all experiments.

Add a clear explanation for EXCBO’s higher starting reward to rule out unfair initialization.

Include MCBO results (even partial) or provide runtime justification with quantitative data.

Standardize reference formatting and include missing citations to heteroscedastic GP and nonlinear ICA literature.

Proofread for typographical errors listed earlier.

**Questions:**

see above

---

> ### Author Response · Authors · 2025-12-04
>
> We thank the reviewer for the constructive comments and for taking the time to evaluate our paper.
>
> **1. Novelty**
>
> We respectfully disagree with the reviewer’s assessment regarding the novelty of this work. Our paper builds on the line of research on counterfactual identification, including ANM (Hoyer et al., NeurIPS 2008), BGM (Nasr-Esfahany et al., ICML 2023), and LSNM (Immer et al., ICML 2023). As demonstrated in Theorem 4.1 and Sections E and F, the proposed DGM constitutes a new family of models that are counterfactually identifiable and can be efficiently implemented using GP. The use of Gaussian mixture models enables learning the recovered exogenous variable distribution, providing a more accurate surrogate of the true data-generating mechanism, as supported by both the paper and our responses. Importantly, the proposed framework is not limited to CBO but applies more broadly to causal inference tasks, including interventions and counterfactual reasoning.
>
> The ideas presented in this paper were motivated by prior research on counterfactual identification (ANM, BGM, LSNM, etc.) and CBO methods (such as MCBO), rather than by the works listed by the reviewer, which do not focus on causality.
>
> **2. Identifiability or robustness, Theoretical depth**
>
> We provide detailed discussions of counterfactual identifiability in Sections E, F, and 4.1. In the revised version, we have strengthened these proofs to improve clarity and rigor.
>
> **3. DGM assumption and missing causal graph**
>
> This paper focuses on a commonly studied setting in CBO where the causal graph is assumed to be known. Extending CBO to simultaneously recover the SCM is highly challenging, and we consider this an important direction for future work. Nonetheless, we include experimental results for a Non-DGM setup in Figure~11, which show that our model continues to outperform existing methods.
>
>  **Exogenous distributions are learned during EXCBO iterations.**
> The recovery of exogenous distributions depends on learning the function $\phi()$ and the decoder $g()$. Therefore, the exogenous distributions must be updated at each iteration $t$, as described in lines 347–348 of the EXCBO algorithm (Algorithm~1).
>
> **4. Experiments**
>
> We have added MCBO results to Figure 4 and Figure 5. All experiments use four random seeds. Furthermore, we report the running time of MCBO in Section D.7. MCBO requires up to 40 times more computation compared with other methods, including EXCBO.

---

### Official Review · Reviewer_XTdz · 2025-11-03

**Soundness:** 3
**Presentation:** 3
**Contribution:** 4
**Rating:** 6
**Confidence:** 4

**Summary:**

The authors built on top of the Sussex MCBO/Aglietti CBO work, incorporating ideas from the exogenous variable learning literature.

They have successfully incorporated previous reviewer feedback, it seems, and improved their presentation and results.

I am happy to increase my score once my three questions are addressed.

**Strengths:**

- The authors present an interesting, novel contribution to the CBO literature, placing it well in context and current literature.
- Exploring the incorporation of EX in CBO is valid and this contribution is therefore relevant to readership.

**Weaknesses:**

- See questions

**Questions:**

1. Why do you not benchmark against CBO by Aglietti et al as a baseline? Is this conceptually incompatible? The code is available and runs without code changes, though requires careful specification of initialisation points, AFAIK.
2. Why do the convergence plots in Figure 6 seem to have different starting points? Presumably, they were initialisation with the identical random sample. Please clarify!
3. “Each figure presents the mean performance over four random seeds” Why can’t you run more seeds? Four seeds seems to be enough for distinguishable error bars, but I am just curious.

Thanks in advance!

---

> ### Author Response · Authors · 2025-12-04
>
> We thank the reviewer for the valuable  comments and for taking the time to evaluate our paper.
>
> **a. Aglietti et al as a baseline**
>
> The CBO framework is based on hard interventions, whereas the baselines used in this paper, as well as EXCBO, rely on soft-intervention approaches. This fundamental difference makes Aglietti et al.\ an unsuitable baseline for direct comparison.
>
> **b. Plot 6**
>
> In Figure 6, all methods use the same initialization dataset size. Under this shared initialization, EXCBO achieves the best performance at the initial step. Smaller initialization datasets can give similar initial reward values, however, too small initial dataset size numbers (<5) are far from real world BO application setups.
>
> **c. More random seeds**
>
> Using more random seeds incurs substantial computational cost due to different $\lambda$ and $\sigma$ combinations in addition to the long running time of MCBO. We found that four random seeds are sufficient to capture the mean and variance.

---

### Official Review · Reviewer_Wa3C · 2025-11-07

**Soundness:** 2
**Presentation:** 2
**Contribution:** 1
**Rating:** 2
**Confidence:** 4

**Summary:**

This paper proposes EXCBO, a causal Bayesian optimization method that learns the distributions of exogenous variables to better model multimodal noise, in the decomposable generation mechasm. The authors incorporated these learned exogenous distributions into the Bayesian optimization process to improve sample efficiency and regret performance.

**Strengths:**

The paper is clearly written and easy to follow, and the proposed method is conceptually straightforward.

**Weaknesses:**

### Weak real-world motivation / failure case.
Without demonstrating or discussing the failure mode of existing works, it's hard to be motivated why we need the proposed method.

### Restrictive design (one action per node).

The method assumes a known mapping where each intervened variable $X_i$ has its own continuous action $A_i$ that directly enters its mechanism. I think it's too restrictive, since in the system, either we want to know the best hard-intervention, or best soft-intervention (find the optimal action values AND their parents). It's hard to come up with a real-world scenario that will be matched with this assumption.

### \tau-SCM appears to rebrand a standard Markovian assumption.

The \tau-SCM is simply an SCM with X = f(Z,U) where Z and U are independent. This is a usual Markovian SCM assumption. I don't see any reasons why authors create a new terminology for already existing notions. Also, it's unclear what \tau stands for.

### Definition 3 (EDS) is not mathematically rigorous.

Defining the encoder/decoder via “a regression model such that E[X] exists and ϕ() can model the conditional mean µϕ() and variance σϕ().", as in Def. 3, is not considered as a mathmatical definition with no rigoursness.

### Extra smoothness assumptions vs prior CBO.

Their identification results require differentiability, whereas prior GP-based CBOs don't assume differentiable structural f in the model statement. Therefore, the claimed “generalization” depends on added smoothness conditions.

### Contradiction: $\hat U=h(Z,X)$ but $\hat U\!\perp\! Z$

Theorem 4.1. mentioned that $\hat U \perp  Z$, while $\hat U$ is a function of $Z$; i.e., $\hat U=h(Z,X)$. I think this is contradictory.

**Questions:**

1. If we infer $U$, we can actually recover the SCM. Then a better optimization algorithm can be found. Please discuss.

2. This model's performance depends on the performance of the encoder models. Please take this account in your errro analysis.

---

> ### Author Response · Authors · 2025-12-04
>
> We thank the reviewer for the valuable comments and for taking the time to review our paper.
>
>
> **1. Real-world motivation**
>
> We conducted experiments on a real-world dataset (Section 7.5) to compare our proposed method with existing baselines. Across these experiments, our approach consistently outperforms UCB, EICF, and MCBO, demonstrating that recovering exogenous variables can substantially improve CBO.
>
> Our work builds on the line of research on counterfactual identification, including ANM (Hoyer et al., NeurIPS 2008), BGM (Nasr-Esfahany et al., ICML 2023), and LSNM (Immer et al., ICML 2023). As shown in Theorem 4.1 and in Sections E and F, the proposed DGM introduces a new family of counterfactually identifiable models that can be efficiently implemented using GP. Leveraging Gaussian mixture models enables us to learn the recovered exogenous variable distribution, yielding a more accurate surrogate of the true data-generating mechanism, as supported by our empirical and theoretical results. Moreover, the framework is not limited to CBO; it extends naturally to broader causal inference tasks, including interventions and counterfactual reasoning.
>
> Finally, we emphasize that multi-modal and non-Gaussian exogenous distributions are prevalent in real-world systems, which further underscores the practical relevance of our approach.
>
> **2. Restrictive design**
>
> We thank the reviewer for raising this concern. Our method can indeed be applied to multi-action models, as demonstrated empirically in Sections 7.4, 7.5, and D.6. The theoretical results and algorithmic design naturally extend to multi-action nodes. We will revise the notations in the paper to make this generality clearer.
>
> **3. Remove $\tau$-SCM definition**
>
> The $\tau$-SCM notation was introduced only to simplify presentation. In the revised version, we have removed this definition and now use \emph{node-SCM} to refer to the single-node SCM model.
>
> **4. Definition 3 (EDS)**
>
> We have improved the rigoursness and clarity of the EDS definition in the revised version.
>
> **5. Contradiction**
>
> There exist many functions of a variable that are independent of the variable itself, even before considering that $h()$ is a function of two variables with paired inputs. A simple single-variable example is $g(X) = \mathbb{E}[X]$, which is a constant and therefore independent of $X$.
>
> For two-variable functions, consider a simple ANM (Hoyer et.al., NeurIPS 2008) example: $X = Z + U$, where $U$ is independent of $Z$. Let $\widehat{U} = h(X, Z) = X - Z = U$. It is straightforward to see that the surrogate $h()$ is independent of $Z$.
>
> In our setting, the encoder $h()$ is a function of both $X$ and $Z$, where $X$ itself is generated from $Z$ and $U$ under the DGM structure. The role of $h()$ is analogous to a normalization function that removes the information of $Z$ from $X$. With sufficient training samples, $h()$ converges to a function that is independent of $Z$, as established in our proof of Theorem 4.1 in Section~E.

---

### Meta-Review · Area_Chair_y1mw · 2026-01-08

**Summary:**

The paper introduces a causal Bayesian optimization method that explicitly learns the distribution of exogenous variables.

- limited real-world motivation and failure cases for existing CBO methods

- lack of novelty, mostly combining known ideas

- restrictive modeling assumptions (single-action nodes, known causal graph)

- lack of theoretical rigor, concerns around  Theorem 4.1

- missing or inconsistent baselines (eg MCBO), limited noise diversity, lack of ablations

- some presentation issues

**Reviewer Concerns:**

Outstanding concerns

- limited novelty

- robustness and identifiability under model misspecification remain unclear

- regret analysis does not account for exogenous-estimation uncertainty

- experimental coverage of more diverse noise types and deeper ablations is limited

- unclear how prior methods concretely fail still

**Reviewer Scores:**

The reviewers were largely negative about the paper, and I doubt the discussion would have significantly changed the situation.

---

### Decision · Program_Chairs · 2026-01-26

Reject